# Enabling Agents to Communicate Entirely in Latent Space

## Abstract

While natural language is the de facto communication medium for LLM-based agents, it presents a fundamental constraint. The process of downsampling rich, internal latent states into discrete tokens inherently limits the depth and nuance of information that can be transmitted, thereby hindering collaborative problem-solving. Inspired by *telepathy*, we propose Interlat (Inter-agent Latent Space Communication), a paradigm that leverages the last hidden states of an LLM as a representation of its mind for direct transmission (termed "latent communication"). An additional compression process further compresses latent communication via entirely latent space reasoning. Experiments demonstrate that Interlat outperforms both fine-tuned chain-of-thought (CoT) prompting and single-agent baselines, promoting more exploratory behavior and enabling genuine utilization of latent information. Further compression not only substantially accelerates inference but also maintains competitive performance through an efficient information-preserving mechanism. We position this work as a feasibility study of entirely latent space inter-agent communication, and our results highlight its potential, offering valuable insights for future research. Our code is available at https://anonymous.4open.science/r/Interlat-9CA4.

## 1 Introduction

Large language model (LLM)-based agentic systems represent a promising and considerably attractive area of contemporary research (Wang et al., 2025; 2024). This interest stems from their ability to orchestrate detailed workflows through natural language that enable agents to interact and collaborate for complex task solving (Qian et al., 2024; Zhang et al., 2024b; Tran et al., 2025). However, although readable by human, natural language introduces constrains on a model's expressive range and can also impose redundant computation: LLMs must down-sample their rich, high-dimensional internal states into discrete tokens, typically exposing only a single linear thought in their message, *i.e.,* a chain of thought (CoT) (Wei et al., 2022) plan to advise another agent (Yu et al., 2024). Furthermore, a large portion of the generated text serves to maintain linguistic coherence rather than to convey essential information (Zhang et al., 2024a). This inefficient paradigm results in ambiguous, lossy inter-agent communication that prevents effective coordination (Chen et al., 2025), which remains a primary cause of task failures in multi-agent systems (Cemri et al., 2025).

To move beyond the limitations of language-based exchange, we explore the idea of communication through the direct transmission of internal representations, enabling more precise and information-preserving interaction. In multi-agent settings, we refer to this as *latent communication*, which instead of inferring others thoughts from words, agents transmit representations that can display their latent thoughts for downstream use, enabling tighter alignment of intentions. While direct sharing is technically and ethically challenging for humans, which is often depicted in fiction (Liu, 2008), *i.e., telepathy,* LLM-based agents spend most of their processing budget in the latent space and naturally produce hidden states throughout their intermediate layers, which can be extracted to support direct, expressive communication. Previous works have attempted to use hidden states for communication; however, these approaches either rely on a one-shot activation graft that struggles to carry multiple reasoning paths. (Ramesh & Li, 2025) or remain constrained by language space, in which hidden states must pair to an already-sampled text trajectory (Tang et al., 2025). Meanwhile, these methods require ad-hoc layer choices, which introduces an extra tuning process.

In this work, we propose Interlat, a novel paradigm for direct inter-agent communication entirely in latent space. Rather than transmitting tokens decoded via the language-model head and embedding layer, Interlat transmits the collected last-layer hidden states for all generated tokens, which we term latent communications, as representations of one agents latent thoughts for another agent. Formally, we frame the agent's ability to communicate in entirely latent space as the ability to differentiate and effectively utilize the rich information contained in the latent communication for the task they are solving. Motivated by the high information density inherent in latent states, inspired by (Hao et al., 2024; Shen et al., 2025; Cheng & Van Durme, 2024), we further train the agent to generate messages in an unconstrained latent space. By learning to generate more information-rich latent states, we successfully compress latent communications into much shorter sequences. This compression yields substantial efficiency improvement while preserving the information needed for downstream tasks.

Experimentally, we focus on a two-agent sender-receiver scenario, which is the fundamental building block of various multi-agent systems, and intentionally avoid orthogonal components such as retrieval, tool use, or multi-round debate orchestration, so as to reduce confounding factors. We evaluate our approach on ALFWorld (Shridhar et al., 2020), a multi-step benchmark requiring planning and execution coordination. Compared to conventional natural language baselines, Interlat achieves performance improvements on both seen and unseen tasks. Analysis reveals that agents utilizing latent communication exhibit more exploratory behavior patterns that lead to higher overall success rates with a genuine understanding of task-relevant latent information rather than superficial pattern matching. Moreover, we demonstrate that latent messages can be compressed to as few as 8 tokens while maintaining competitive performance, achieving up to a $24\times$ reduction in communication latency. Further analysis of the output probability distribution after compression reveals how task-critical information is effectively preserved.

## 2 RELATED WORK

**Latent Reasoning in LLMs.** A growing line of work shifts reasoning from the language space to the latent space, replacing explicit CoT traces with multi-step computation in continuous representations to bypass the bandwidth and efficiency constraints of text ($\approx 15$ bits/token vs. $\approx 40k$ bits/hidden-state) (Zhu et al., 2025b). To increase available compute at inference time, (Goyal et al., 2023) introduces learnable pause tokens that delay the generation of the final answer, while (Pfau et al., 2024) employs filler tokens to scaffold intermediate computations that would be infeasible without generating tokens. Beyond token scheduling, (Liu et al., 2024) proposes a latent coprocessor operating directly on the transformer KV cache to improve performance. Another line of work (Hao et al., 2024; Shen et al., 2025; Cheng & Van Durme, 2024) enables the model to reason in latent space by feeding the last hidden state back as the next input embedding, enabling the model to explore multiple reasoning paths in parallel, akin to a breadth-first search. (Bae et al., 2024; Gao et al., 2024; Geiping et al., 2025) decouple input encoding, iterative reasoning, and output decoding, making the computation more modular and interpretable. Building upon these advantages of latent space for reasoning, our work shifts focus from single-model latent reasoning to enabling inter-agent communication and task solving in latent space.

**Multi-agent Communication.** LLM-based agent systems typically orchestrate in natural language (Qian et al., 2024; Zhu et al., 2025a; Wang et al., 2024). Although readable by humans, natural language introduces constraints on a model's expressive range and can also impose redundant computation. Classical studies in emergent communication (Lazaridou et al., 2016; 2018; Tucker et al., 2022; 2021) have shown that small neural agents can learn to develop symbolic or low-dimensional continuous protocols in referential games or cooperative reinforcement learning settings, illuminating how communication efficiency and semantic structure may arise. However, these frameworks operate on an explicit, separate communication channel (discrete tokens or bottleneck vectors) that is learned from scratch and detached from the agents internal reasoning states. By contrast, recent LLM-based methods shift communication into richer representational forms: Pham et al. (2023) replaces sampled tokens with probability-weighted tokenizer embeddings, preserving more of the models belief distribution and improving debate performance, but it still communicates surface-level final-layer distributions, overlooking deeper, more informative, and more valuable hidden representations. Ramesh & Li (2025) blends hidden states between agents, yielding accuracy gains with less compute than natural-language messages; however, it operates as a single hidden-state graft within

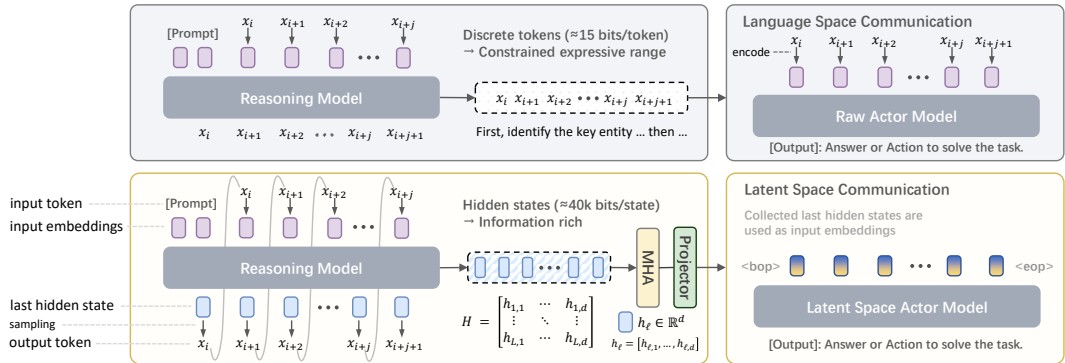

Figure 1: A comparison of Interlat with conventional language-space communication. In language space, an agent transmits a discrete token sequence $[x_i, x_{i+1}, \ldots, x_{i+j+1}]$ (*e.g.,* a CoT plan) to another. In Interlat, the model leverages its last hidden states as a representation of its internal mind state, processed by a communication adapter, and then transmits them directly to the other agent, enabling communication entirely in latent space with higher expressive capacity.

a pass rather than a temporally structured latent sequence. Tang et al. (2025) preserves richer information by recording per-token state deltas at selected layers on the sender and adding them at the corresponding layer positions on the receiver while transmitting text, but requires model-specific layer selection and language-space text. Building on these valuable insights, we transmit a sequence of last hidden states directly between agents and apply a compression process to enable information-rich, language-free, and efficient communication.

## 3 INTERLAT

In this section, we formalize how to extract an agents states as the representation of its "mind" for inter-agent latent space communication. Let $x = (x_1, \ldots, x_T)$ with a prompt $x_{1:m}$ and a completion $y = (y_1, \ldots, y_L)$ such that $y_\ell = x_{m+\ell}$ and $L = T - m$. For each decoding step $\ell = 1, \ldots, L$, define

$$
\begin{aligned}
h_\ell &= \text{Transformer}(x_{\leq m+\ell-1})\,[\,m+\ell-1\,], \\
H &= [\,h_1,\, h_2,\, \ldots,\, h_L\,],
\end{aligned}
\tag{1}
$$

where $h_\ell \in \mathbb{R}^d$ is the last hidden state immediately before predicting $y_\ell$ (i.e., at position $m + \ell - 1$ in the full sequence). $H \in \mathbb{R}^{L \times d}$ collects these last-layer hidden states for the completion region.

### 3.1 LATENT COMMUNICATION

Our Interlat removes natural language constraints by letting agents transmit their thoughts by directly passing the collected last hidden states, which we termed **latent communication**. As shown in Figure 1, this transmission occurs at the end of an agent's message generation process. Special tokens, $x_i = $ <bop> and $x_j = $ <eop>, are added to mark the beginning and the end of the latent communications. Consider an agent $\mathcal{M}_i$ solving a task $\mathcal{T} = \{x_1, \ldots, x_m\}$. Upon receiving a latent communication $H = \{h_1, h_2, \ldots, h_L\}$ from another agent, it forms its input embedding as:

$$
E = [\,e(x_1), e(x_2), \ldots, e(x_i), h_1, h_2, \ldots, h_L, e(x_j)\,],
$$

where $e(\cdot)$ is the token embedding function. This inference process is analogous to standard language space multi-agent systems, except that it feeds hidden states between agents. The latent communications have been processed by a trainable light-weight self-attention and a projection layer as a communication adapter for magnitude rescaling and helping the agent better interpret the latent meaning within these latent representations. For brevity, we may refer to latent communications as *latents* where unambiguous.

### 3.2 TRAINING PROCEDURE

In this work, we consider two agents: a *reasoning* agent as a sender that produces a task-specific plan together with its final-layer latent states, and an *actor* agent as a receiver that consumes this information to generate actions to solve tasks, which is the basic unit for sophisticated systems.

Let $Y_t$ denote the next token at supervised position $t \in S$, and $C_t$ the decoder prefix including task tokens and past outputs up to $t$. We encourage the actor to understand and utilize $H$ by directly maximizing a conditional distributional separation regulated objective by supervised fine-tuning (SFT):

$$\mathcal{L}_{\text{total}} = \underbrace{\mathcal{L}_{\text{task}}}_{\text{next-token CE}} + \lambda_S \underbrace{\mathcal{L}_{\text{sep}}}_{\text{JS separation loss}} + \lambda_A \underbrace{\mathcal{L}_{\text{align}}}_{\text{plan-alignment}},$$

where $\lambda_S, \lambda_A > 0$, and $\mathcal{L}_{\text{task}}$ is the standard cross-entropy loss that ensures the model produces accurate and coherent responses based on the given task.

**Conditional mind separation.** In this work, we compare the full conditional distributions $p_\theta$ of the softmax of logits under matched ($H$) and mismatched latents ($\tilde{H}$), where $\tilde{H}$ is a latent communication from another example in the same training batch (*i.e.*, a latents from a different task). We minimize a weighted Jensen-Shannon divergence (Lin, 2002):

$$\mathcal{L}_{\text{sep}} = -\frac{1}{|S|} \sum_{t \in S} \text{JS}\big(p_\theta(\cdot \mid C_t, H), \ p_\theta(\cdot \mid C_t, \tilde{H})\big).$$

This objective separates matched from mismatched conditional distributions, providing a robust and efficient training signal that encourages the agent to understand and leverage the latent information.

**Plan-aligned regulation.** Maximizing the JS separation encourages the actor to respond to the $H$, but it also opens a failure mode where the model can game the objective by moving probability toward idiosyncratic tokens that increase the divergence while harming task utility. To prevent such degenerate solutions, we regularize predictions conditioned on $H$ with those conditioned on the language space plan ($P$). Specifically, $P$ denotes a natural-language sequence describing the reasoning steps for communication (*e.g.*, a CoT plan), that is generated by the same instruction-tuned model during the autoregressive generation of $H$ shown in Figure 1. We use an instruction-tuned model to ensure that the plan is semantically coherent, well-structured, and aligned with the task-solving intention. Let $p_{\text{plan}}(\cdot \mid C_t, P)$ denote the distribution when only the plan is provided:

$$\mathcal{L}_{\text{align}} = \frac{1}{|S|} \sum_{t \in S} \Big[ \beta \, \text{KL}\big(p_\theta(\cdot \mid C_t, H) \,\|\, p_{\text{plan}}(\cdot \mid C_t, P)\big) + \alpha\big(1 - \cos\big(\ell_\theta(C_t, H), \ell_{\text{plan}}(C_t, P)\big)\big) \Big],$$

where $\ell_\theta$ and $\ell_{\text{plan}}$ are the corresponding normalized logit vectors. All divergences and cosines are computed at supervised positions, probabilities are taken as softmax of logits.

**Curriculum Learning.** We employ a curriculum learning strategy to progressively train agents to understand the rich information embedded in latent states for communication. Inspired by (Su et al., 2025), we employ a stochastic replacement strategy that substitutes portions of latents with corresponding text embeddings in a left-to-right manner. Specifically, let $x_{1:L}$ be the input communication tokens from a reasoning model, with $e(x_{1:L}) = \{e_1, e_2, \ldots, e_L\}$ as their embeddings and $H = \{h_1, h_2, \ldots, h_L\}$ as the latent states obtained during the autoregressive generation of $x_{1:L}$. For each training instance, we uniformly sample a replacement rate $r$ from the set $R = \{0, 0.1, \ldots, 1.0\}$. We then replace the first $\lfloor r \cdot L \rfloor$ latent states with their token embeddings, forming a mixed input sequence:

$$H^{(r)} = \underbrace{e_1, \ldots, e_{\lfloor r \cdot L \rfloor}}_{\text{token embeddings}} \oplus \underbrace{h_{\lfloor r \cdot L \rfloor + 1}, \ldots, h_L}_{\text{latent states}}.$$

This method enhances training efficiency while achieving strong model performance. By integrating this curriculum, the final training objective is to minimize the following expectation over the replacement rate:

$$\mathcal{L}_{\text{total}}(\theta) = \mathbb{E}_{r \sim p_R} \Bigg[ \frac{1}{|S|} \sum_{t \in S} \Bigg( \underbrace{-\log p_\theta(y_t \mid C_t, H^{(r)})}_{\mathcal{L}_{\text{task}}(t)} - \lambda_I \underbrace{\text{JS}\Big(p_\theta(\cdot \mid C_t, H^{(r)}), p_\theta(\cdot \mid C_t, \tilde{H})\Big)}_{\mathcal{L}_{\text{sep}}}$$

$$+ \lambda_A \underbrace{\Big[ \beta \, \text{KL}\Big(p_\theta(\cdot \mid C_t, H^{(r)}) \,\|\, p_{\text{plan}}(\cdot \mid C_t, P)\Big) + \alpha\Big(1 - \cos\big(\ell_\theta(C_t, H^{(r)}), \ell_{\text{plan}}(C_t, P)\big)\Big) \Big]}_{\mathcal{L}_{\text{align}}(t)} \Bigg) \Bigg].$$

### 3.3 INFORMATION COMPRESSION

While full-length latent communications $H_L \in \mathbb{R}^{L \times d}$ are rich in information, their length $L$ (often dozens to hundreds) limits communication efficiency. To address this, we train a compact reasoning model $M_\phi$ to generate much shorter latent sequences $H_K \in \mathbb{R}^{K \times d}$ ($K \ll L$, e.g., $K = 8$) while preserve task-critical information for the actor. Rather than simply truncating $H_L$, we let $M_\phi$ reason entirely in latent space by feeding its own last hidden state back as the next input embedding, following a fully differentiable autoregressive loop:

$$\langle M_\phi(E_i) \to h_i, \quad E_{i+1} = E_i \oplus \text{Proj}(h_i)\rangle,$$

where $\text{Proj}(\cdot)$ is a small linear projection. This enables end-to-end training of compact yet expressive latent messages without decoding to tokens, preserving rich information continuously within the generation process. As shown in Figure 2, we freeze an instruction-tuned model as a teacher that generates full-length latents $H_L$ in the previous actor model training process, which serve as a reference for what constitutes a useful, task-aligned latent message. We keep the actor model and communication adapter frozen during this stage, ensuring that compression is trained solely to match the actors expected input distribution.

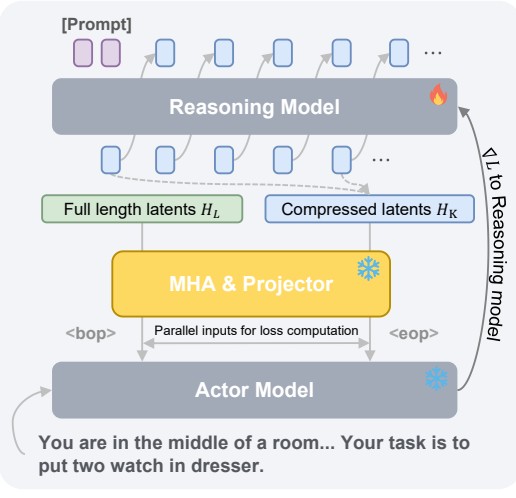

Figure 2: Training the reasoning model with frozen-actor supervision.

To ensure that compression preserves task-relevant information, we jointly optimize a composite objective comprising three components: (i) a task-utility loss that enforces correct predictions from the frozen actor using the compressed latents $H_K$; (ii) an uncertainty-weighted KL agreement that aligns the actors output distribution under $H_K$ with that under the full-length latents $H_L$, prioritizing positions where $H_L$ most reduces predictive uncertainty; and (iii) a cosine alignment loss that maintains geometric consistency between the actor-side feature representations induced by $H_K$ and $H_L$. Formally, the compression objective is:

$$\mathcal{L}_{\text{compress}} = \lambda_{\text{task}}\mathcal{L}_{\text{task}} + \lambda_{\text{pref}}\mathcal{L}_{\text{pref}} + \lambda_{\text{geom}}\mathcal{L}_{\text{geom}},$$

where $\mathcal{L}_{\text{pref}} = \mathbb{E}_{t \in S}[w_t \cdot \text{KL}(p_t^{(L)} \| p_t^{(K)})]$, and $\mathcal{L}_{\text{geom}} = 1 - \cos(\bar{z}^{(K)}, \bar{z}^{(L)})$. Here, $w_t = \max(H(p_t^{(B)}) - H(p_t^{(L)}), 0)$ reflects the uncertainty reduction from $H_L$, and $\bar{z}^{(\cdot)}$ denotes the mean actor-side latent features. This design encourages $M_\phi$ to retain both functional behavior and structural semantics of the original message while discarding redundancy, effectively learning an information-preserving bottleneck in latent space. A full derivation is provided in Appendix B.

## 4 EXPERIMENT

**Implementation Details.** We evaluate our approach on the ALFWorld (Shridhar et al., 2020), a multi-step embodied reasoning benchmark that requires agents to plan and act within a simulated household environment (We give a detailed description and rationale for using ALFWorld in Appendix D.). We adopt the official split with 3119 training tasks, 140 validation tasks, and 134 test tasks. Episodes are capped at 20 environment steps; Success is 1 if the goal state is reached within the budget and 0 otherwise. All models are trained offline on ALFWorld trajectory data that include task descriptions, step-by-step thoughts, and executed actions from (Song et al., 2024), environment rewards are only used for evaluation, not for training. We provide a training example in Appendix J. The Qwen2.5-7B-Base and Qwen2.5-0.5B-Base models (Yang et al., 2024) serve as the actor agents, while their instruction-tuned counterparts (Qwen2.5-7B-Instruct and Qwen2.5-0.5B-Instruct) are used to generate Chain-of-Thought (CoT) plans and uncompressed latent communications. Training is conducted with mixed-precision (bfloat16), FlashAttention-2 (Dao, 2023), gradient checkpointing, and DeepSpeed ZeRO-2 (Rajbhandari et al., 2020) across 8 NVIDIA A100-80G GPUs. A 5% validation split is used for model selection, and batch-level early stopping is applied to the reasoning model. For compression experiments, a separate Qwen2.5-7B-Base model is trained as the reasoning agent with the actor kept frozen. All results are averaged over three independent runs for statistical robustness. Further implementation details are available at Appendix C.

| Method | Qwen2.5-7B-Base | | | | Qwen2.5-0.5B-Base | | | |
|---|---|---|---|---|---|---|---|---|
| | Seen | Steps | UnSeen | Steps | Seen | Steps | UnSeen | Steps |
| **Interlat** | | | | | | | | |
| Ours | **70.48** | 9.41/12.54 | **65.42** | 9.86/13.37 | **61.19** | 10.55/14.22 | **57.46** | 9.38/13.90 |
| Text | 64.29 | 8.76/12.77 | 62.44 | 9.79/13.63 | 54.52 | 9.50/14.28 | 47.26 | 9.70/15.13 |
| No-Comm | 62.14 | 10.19/13.90 | 62.19 | 10.23/13.92 | 50.48 | 8.23/14.06 | 44.03 | 9.10/15.20 |
| **Baselines** | | | | | | | | |
| CoT (full) | 67.14 | 8.15/12.04 | 64.93 | 9.02/12.87 | 57.86 | 8.30/13.23 | 50.75 | 8.94/14.39 |
| No-CoT | 65.71 | 8.23/12.27 | 62.69 | 9.15/13.20 | 57.14 | 8.96/13.69 | 50.25 | 9.80/14.87 |
| **Variants** | | | | | | | | |
| CrossTask | 61.43 | 8.42/12.89 | 61.94 | 9.51/13.50 | 53.57 | 9.40/14.32 | 47.01 | 10.06/15.33 |
| Noised | | | | | | | | |
|   CovNoise-0.5× | 64.29 | 8.54/12.63 | 60.95 | 8.71/13.12 | 53.33 | 8.80/14.03 | 46.77 | 9.64/15.16 |
|   CovNoise-1.0× | 63.81 | 8.66/12.76 | 63.68 | 8.72/12.82 | 53.10 | 8.96/14.14 | 44.53 | 9.68/15.40 |
|   WhiteNoise | 61.90 | 8.65/12.97 | 61.19 | 9.32/13.46 | 57.38 | 8.00/13.11 | 57.21 | 9.18/13.81 |
| CovGauss-0µ | 60.00 | 8.79/13.27 | 61.94 | 9.59/13.55 | 13.81 | 11.25/18.79 | 13.18 | 12.93/19.07 |
| CovGauss-µ | 65.71 | 8.58/12.50 | 64.93 | 8.63/12.62 | 44.52 | 9.21/15.20 | 34.33 | 10.19/16.63 |
| RandomRot | 57.86 | 8.43/13.31 | 63.68 | 9.37/13.23 | 59.05 | 8.24/13.06 | 51.99 | 9.12/14.34 |

Table 1: Performance of different methods and variants on seen and unseen tasks. Higher success rates indicate stronger inter-agent collaboration and task-solving ability. Steps reports average steps on successful tasks and average steps over all tasks, separated by a slash.

**Baselines and settings in Interlat.** In this work, we seek to study the feasibility of agents communicating *entirely* in latent space; accordingly, we consider the following baselines: (1) **CoT (full).** We use complete Chain-of-Thought (CoT) traces produced by a related instruction-tuned model (Qwen2.5-7B-Instruct and Qwen2.5-0.5B-Instruct) to perform full-parameter supervised fine-tuning. In inference, the model receives a complete CoT plan before generating answers. (2) **No-CoT.** The model is trained to produce the final answer directly, without receiving any plan from other agents.

We also evaluate some variants of our method: (1) **Text.** Instead of latent communication, we feed the corresponding CoT plan to the actor. (2) **No-Comm.** We remove any communication from the actor's input. (3) **CrossTask.** We replace the current tasks latents with one sampled from a different task. (4) **Noised.** We add perturbations to the latent communication $H$: (a) **CovNoise-0.5×/1.0×**: covariance-shaped noise $\varepsilon_t \sim \mathcal{N}(0, \hat{\Sigma})$ with optional strength $\lambda \in \{0.5, 1.0\}$, where $\hat{\Sigma}$ is the sample covariance of the original $H$; (b) **WhiteNoise**: a control drawn from $\mathcal{N}(0, I)$ with the same length. (5) **CovGauss.** The latents is replaced by Gaussian samples that match the original's mean and covariance but lack its higher-order structure. (6) **RandomRot.** A random orthogonal rotation is applied to the latents, perfectly preserving its mean and covariance while scrambling its higher-order structure. Further implementation details are available at Appendix C.

## 4.1 MAIN RESULT

Table 1 presents a comprehensive comparative analysis of the Interlat framework against other methods. Latent communications effectively enhance agents' task-solving ability, as shown by the consistent improvement over fine-tuned single agent and agents trained to communicate in natural language. We describe several key findings from the experiment as follows.

**Latent Communication Prompts Agent Exploration.** Beyond accuracy improvements, removing the constraints of language space yields longer yet more successful trajectories for agents with latent communication. Even without explicitly training exploration policies, by effectively leveraging other agents' multiple plausible reasoning paths in latent communication, it naturally engages in more thorough exploratory behavior patterns. This phenomenon suggests an enhanced environmental understanding rather than random exploration, which in turn leads to higher overall task success rates than agents with a natural language plan with fewer steps. A deeper analysis of the relationship between step counts and success rates is provided in the Appendix E.

**Actor Agent Actually Comprehend Latent Communication.** With dense and implicit information in the latent space, we test whether the trained actor agent genuinely exploits the latent information carried by latent communication, rather than matching superficial distributional cues. As shown in Table 1, replacing task-specific latents with cross-task mismatch latents causes a substantial accuracy drop. Although these foreign messages share similar global statistics, they encode rea-

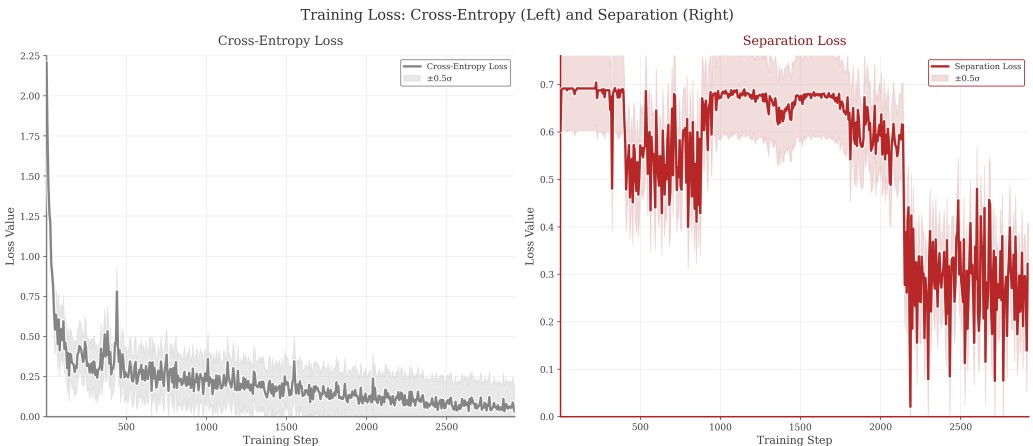

Figure 3: Training dynamics of the cross-entropy loss and separation loss: an initial plateau near 0.69 indicates no separation between matched/mismatched latents, followed by a sharp drop after $\sim$ 2.2k steps, marking the models aha moment in exploiting task-relevant latent information.

soning paths irrelevant to the current task. Accuracy declines further when we substitute covariance-matched Gaussian surrogates or apply random orthogonal rotations to latents, variances that preserve first- and second-order moments while disrupting structure. Performance also drops under additive noise to the original latents, which hampers the extraction of the information. White noise, which contains no structured content, also impairs performance. Overall, these variances indicate that successful execution depends on understanding both the meaningful, task-specific information and the inherent structure of the latent communication, which represents constructed reasoning paths.

**An 'Aha' Moment in Understanding Latent Information.** Learning to interpret latent communication aligns with an LLM's inherent processing modality, as these models are pre-trained on natural language but internally process information using hidden states. Through curriculum learning and tailored loss functions, we guide LLMs to gradually comprehend latent information. As shown in Figure 3, the separation loss reveals a distinct two-phase learning curve. For the first 2,200 steps, the loss plateaus near 0.69, which is the maximum Jensen-Shannon divergence, indicating an inability to distinguish task-specific from cross-batch latent messages. Around step 2,200, however, the loss drops sharply, indicating a qualitative leap reflects the emergence of a genuine understanding and leveraging of latent information for task solving. Notably, using cross-batch messages as negatives poses a more significant challenge than using random noise, as they are coherent yet task-irrelevant. The models ability to achieve eventual discrimination underscores its move beyond superficial patterns toward meaningful latent communication.

## 4.2 COMPRESSION

**Compression Analysis.** Theoretically, with a much higher expressive range, latent communications can encode rich information in far fewer positions than natural-language tokens. To quantify their compression capacity, we evaluate two distinct settings: **i) Training-free**: We directly use the off-the-shelf Qwen2.5-7B-Instruct model to generate full-length latent communications, which are used for actor model training. And then truncate or uniformly subsample them to shorter lengths without any additional training. This tests the intrinsic compressibility of raw latent communication. **ii) Trained**: We use a compression-trained Qwen2.5-7B-Base reasoning model that is explicitly optimized to autoregressively generate compact latent sequences of target length $K$ in a complete latent space while preserving task-relevant information under supervision from a frozen actor.

As shown in Table 2, under the training-free setting, Interlat achieves optimal performance at 50% compression with a success rate of 72.14%, even surpassing the uncompressed baseline. This suggests that moderate compression preserves essential information while reducing computational overhead. However, performance becomes volatile at extreme compression ratios around 20% compression, indicating sensitivity to extreme compression ratios. When hidden states are completely removed, accuracy drops to 62.14%, confirming their critical role in information transmission. Un-

| Ratio | Seen | Unseen | Time |
|---|---|---|---|
| **Untrained** | | | |
| Full | $\underline{70.48}_{\pm 1.01}$ | $\underline{65.42}_{\pm 0.87}$ | 9.19s |
| 90% | $68.57_{\pm 1.63}$ | $\mathbf{67.16}_{\pm 1.97}$ | - |
| 80% | $68.10_{\pm 1.83}$ | $61.69_{\pm 1.43}$ | - |
| 70% | $67.14_{\pm 1.82}$ | $63.43_{\pm 2.24}$ | - |
| 60% | $66.43_{\pm 1.63}$ | $59.20_{\pm 3.69}$ | - |
| 50% | $\mathbf{72.14}_{\pm 1.48}$ | $61.19_{\pm 2.84}$ | - |
| 40% | $66.90_{\pm 2.31}$ | $59.95_{\pm 2.64}$ | - |
| 30% | $65.95_{\pm 2.12}$ | $62.19_{\pm 1.58}$ | - |
| 20% | $67.86_{\pm 3.23}$ | $61.44_{\pm 1.58}$ | - |
| 10% | $67.86_{\pm 2.12}$ | $62.44_{\pm 2.64}$ | - |
| 5% | $64.29_{\pm 1.12}$ | $60.95_{\pm 1.35}$ | - |
| 0% | $62.14_{\pm 2.01}$ | $62.14_{\pm 2.32}$ | - |

| Ratio | Seen | Unseen | Time |
|---|---|---|---|
| **Untrained** | | | |
| 128L | $64.55_{\pm 2.26}$ | $60.25_{\pm 2.06}$ | 3.55s |
| 64L | $66.23_{\pm 1.95}$ | $61.53_{\pm 4.32}$ | 1.83s |
| 32L | $63.57_{\pm 2.01}$ | $60.18_{\pm 3.58}$ | 1.03s |
| 16L | $64.29_{\pm 1.34}$ | $60.00_{\pm 3.01}$ | 0.62s |
| 8L | $64.00_{\pm 2.18}$ | $57.46_{\pm 2.69}$ | 0.39s |
| **Trained** | | | |
| 128L | $\mathbf{68.10}_{\pm 1.93}$ | $\mathbf{62.94}_{\pm 2.03}$ | 2.25s |
| 64L | $\underline{67.14}_{\pm 1.56}$ | $\underline{61.94}_{\pm 2.13}$ | 1.16s |
| 32L | $66.90_{\pm 1.46}$ | $61.94_{\pm 2.56}$ | 0.60s |
| 16L | $66.43_{\pm 2.05}$ | $61.69_{\pm 2.56}$ | 0.33s |
| 8L | $66.43_{\pm 1.22}$ | $60.45_{\pm 2.23}$ | 0.20s |

Table 2: Result of compression on seen and unseen tasks. **Left:** training-free sweep over retained ratio $R$ from full to zero. **Right:** varying the length of transmitted latents $L \in \{8, 16, 32, 64, 128\}$ for training-free and trained reasoning models. Time denotes end-to-end latency (s) of the message generation process. The top scores are in **bold**, with the second-highest underlined.

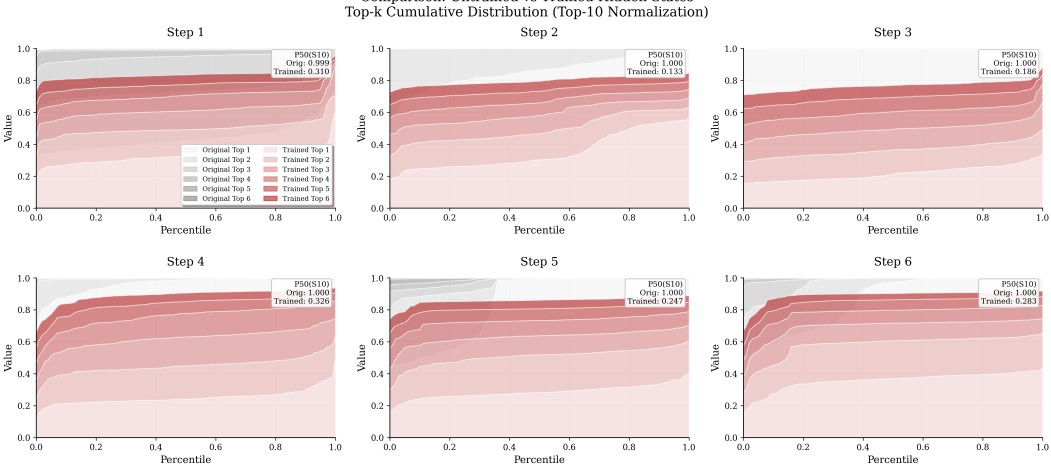

Figure 4: Analysis of parallelism in latent communication across the first six steps. The red denotes latents from the trained model, and the gray is the untrained model. Trained latents retain stable vertical gaps between successive Top-$k$ bands and exhibit markedly lower $P_{50}(S_{10})$, indicating persistent parallelism, whereas the untrained latents progressively collapse toward Top-1.

der the training setting, we apply various compression rates to the reasoning model, generating latent communications of 8, 16, 32, 64, and 128 hidden states ($\approx 1.8\%$, $3.6\%$, $7.2\%$, $14.4\%$, and $28.8\%$ of the full sequence length, respectively) for the action agent. This compression training process yields more stable performance and higher success rates across various compression levels, demonstrating the model's ability to learn efficient, compressed representations.

Furthermore, compression markedly improves wall-clock efficiency. By generating far fewer hidden states for inter-agent communication, the average end-to-end latency drops from 9.19 s at full length to 0.39 s with an 8-step latents, a nearly $24\times$ speed-up. Although with a lightweight bridge module, the trained reasoning agent further reduces the generation runtime to 0.20s by largely eliminating the decode-re-encode overhead inherent in token-based communication.

**Why compression is effective.** To understand the mechanisms that make compression effective, we first measure its impact on predictive uncertainty by sweeping the communication rate $R \in [0, 1]$ and computing the cross-entropy (CE, in bits/hidden state). Define $\mathrm{CE}_{\mathrm{full}}$ as the CE under full communication, and $\mathrm{CE}_{\mathrm{comp}}(R)$ as the CE at rate $R$. We report the task-averaged

| Actor | Seen | Unseen |
|---|---|---|
| Ours Full | $\mathbf{70.48}_{\pm 1.01}$ | $\mathbf{65.42}_{\pm 0.87}$ |
| w/o curri | $33.10_{\pm 2.97}$ | $20.65_{\pm 2.15}$ |
| w/o $\mathcal{L}_{\text{sep}}$ | $\underline{58.81}_{\pm 1.41}$ | $\underline{60.70}_{\pm 5.50}$ |
| w/o $\mathcal{L}_{\text{align}}$ | $56.90_{\pm 1.41}$ | $53.98_{\pm 3.35}$ |
| w/o adapter | $4.05_{\pm 1.70}$ | $4.48_{\pm 1.31}$ |

| Reasoning | Seen | Unseen |
|---|---|---|
| Ours Full | $\mathbf{68.10}_{\pm 1.93}$ | $\underline{62.94}_{\pm 2.03}$ |
| w/o $\mathcal{L}_{\text{task}}$ | $65.71_{\pm 1.43}$ | $\mathbf{63.18}_{\pm 3.47}$ |
| w/o $\mathcal{L}_{\text{pref}}$ | $64.76_{\pm 2.97}$ | $60.20_{\pm 3.13}$ |
| w/o $\mathcal{L}_{\text{geom}}$ | $64.05_{\pm 3.55}$ | $59.45_{\pm 3.01}$ |

Table 3: Ablation of training components. See Appendix E for full results and analysis.

relative change $\Delta\text{CE}\%(R) = 100 \times \frac{\text{CE}_{\text{comp}}(R) - \text{CE}_{\text{full}}}{\text{CE}_{\text{full}}}$. Figure 5 reveals several key insights: (1) $\Delta\text{CE}\%$ decreases monotonically with increasing $R$, exhibiting a plateau region from approximately 30% to 75% bits/hidden state, which corresponds to the range of optimal performance; (2) the trained latent communication curve consistently outperforms the training-free curve across all compression rates, with the vertical gap reaching a maximum of approximately 11 percentage points; (3) the trained reasoning agent achieves substantial computational savings-up to 131,421.5 rates (42.5% reduction) at optimal compression points while maintaining competitive performance.

We further examine how information is preserved under compression. The probability distribution serves as the models implicit *information parallelism budget*, estimating how many viable possibilities are preserved at each step. Figure 4 shows the parallelism of the reasoning agent's latent communication by examining the first six hidden states, each representing a distinct reasoning step. For each step, we compute the output probability distribution from the language model (LM) head and plot the cumulative probability mass of the top-$k$ tokens ($k = 1 \ldots 6$) across communication percentiles and renormalize within top-10 steps. Steps in latents from the trained reasoning agent exhibits stable vertical gaps between successive top-$k$ curves across steps, indicating persistent parallelism rather than the progressive convergence observed in the untrained model. Furthermore, we measure head coverage using P50($S_{10}$), defined

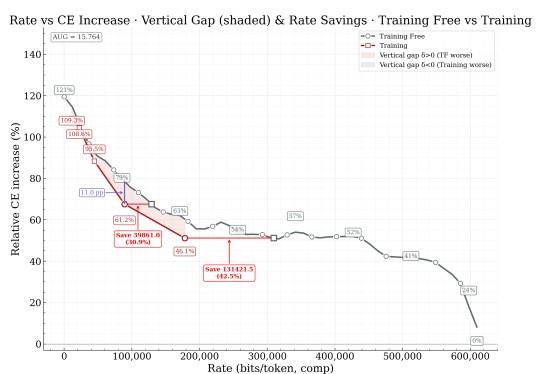

Figure 5: The task-averaged relative change $\Delta\text{CE}$ and the relative saving rates before and after the compression training process.

as the median probability mass of the top-10 tokens ($S_{10} = \sum_{i=1}^{10} p_{(i)}$) across the communication. The trained model shows a significantly lower P50($S_{10}$), suggesting that the trained model preserves a broader set of plausible reasoning paths within a compressed latent representation. These results demonstrate that the trained model sustains diverse parallel exploration over multiple reasoning steps, avoiding premature collapse into a single hypothesis in communications. This capability enables the model to retain richer information for downstream tasks. A detailed results with extended step-wise results is provided in Appendix H.

## 4.3 ABLATION STUDIES

To assess the contribution of each training component, we performed a systematic ablation study on both the actor and reasoning models. The results, summarized in Table 3, confirm that each component is crucial for achieving optimal performance.

For the actor model, removing curriculum learning forces the model to interpret latent communications from scratch, leading to extremely unstable training dynamics and severely degraded comprehension, as illustrated in Figure 6. Removing the separation loss induces shortcut behavior; the model learns to ignore the latent communication and rely only on the textual task prompt, causing performance to regress toward the single agent baseline. Removing the communication adapter causes the largest drop; despite generating fluent and coherent responses, the model fails to complete tasks, underscoring the adapters role in bridging the agents latent spaces and enabling interpretation of latent communications.

For the reasoning model, which is trained to generate compressed latents, we ablated its three core loss functions with compressed target length $K = 128$. The most critical component is the latent direction alignment loss ($\mathcal{L}_{\text{geom}}$). This highlights the importance of maintaining geometric consistency between the compressed latents and the uncompressed ones. The uncertainty-weighted agreement loss ($\mathcal{L}_{\text{pref}}$) is also vital, as removing it significantly impairs the model's ability to produce latents that elicit the correct behavior from the actor. Interestingly, removing the actor's cross-entropy loss ($\mathcal{L}_{\text{task}}$) slightly improves performance on unseen tasks, from 62.05 to 62.90, suggesting that while directly optimizing for the frozen actor's outputs is beneficial for in-distribution tasks, it may cause minor overfit-

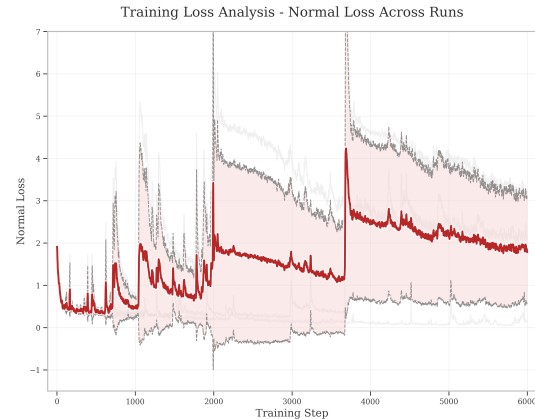

Figure 6: Training dynamics of the cross entropy loss when removing curriculum learning.

ting that slightly hinders generalization. We leave a deeper investigation into this trade-off to future work.

## 5 Qualitative Analysis of Latent Communication via t-SNE

To qualitatively assess the semantic structure of latent communications, we visualize all 3,119 mean-pooled latent communication via t-SNE shown in Figure 7). Clear clustering by task template (*e.g.,* `pick_and_place`) provided by official Alfworld benchmark demonstrates that latent communications encode task-specific semantic information rather than random noise, forming the foundation that enables the actor to effectively interpret and leverage reasoning patterns of varying procedural complexity. Furthermore, intra-cluster dispersion reveals that even within a single template, the representations preserve object-receptacle-specific variations, reflecting diverse latent reasoning paths. This confirms that Interlat transmits structured, actionable information without relying on natural language.

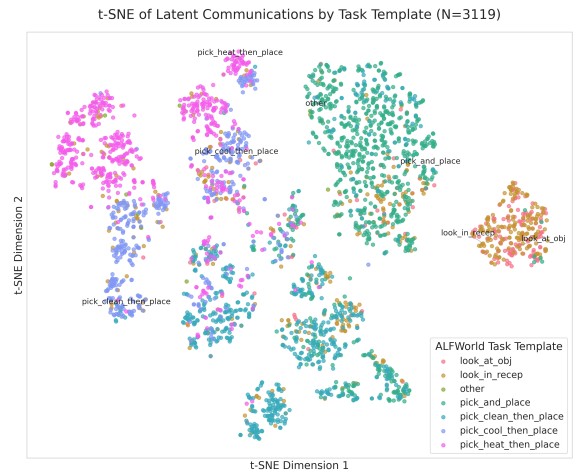

Figure 7: t-SNE visualization of latent communications grouped by ALFWorld task template.

## 6 Conclusion

In this paper, we presented Interlat, a novel paradigm for inter-agent communication entirely in latent space. Through extensive experiments, we demonstrated that Interlat successfully enhances agents' task-solving ability and communication efficiency over language-based methods by effectively utilizing task-related latent information. Analysis also highlighted how latent messages can be highly compressed while retaining performance by preserving diverse, parallel reasoning paths. Future work can further explore the integration of models from different families and leveraging hidden states from multiple layers to enable richer, more insightful communication. Another promising direction is joint training of more agents to communicate in latent space, which has the potential to scale our method. We anticipate these findings will encourage broader study into latent space communication and contribute to the development of more advanced multi-agent systems.

# 7 ETHICS STATEMENT

No human participants, crowdsourcing, or personally identifiable information (PII) were involved in this research. All experiments were conducted within a simulated environment using standard dataset splits.

Our study focuses on inter-agent communication in latent space, utilizing the last hidden states and their compressed variants. A potential theoretical risk is that such latent communication could be exploited to circumvent language-based safety mechanisms. To mitigate this concern to the greatest extent possible, we neither trained on nor evaluated any harmful instructions, and no harmful actions occurred during our experiments. Furthermore, to promote transparency, we analyze the internal probability distribution of latent communications in Section 4.2, which provides a clearer understanding of the information being transmitted.

# 8 REPRODUCIBILITY STATEMENT

To facilitate reproducibility, we provide an anonymous repository link at the end of our abstract containing complete training and inference code and configuration files [1]. The core method and training objective are specified in the method section. Implementation details and evaluation protocols containing models, baselines, metrics, and stopping criteria are in both the Experiment section and the Appendix C. Appendix J also includes samples from our dataset and the templates used for agent training.

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

## Appendix

The supplementary information accompanying the main paper provides additional data, explanations, and details.

## A  LLM USAGE

ChatGPT[2] was used purely with the language of the paper during the writing process, including spell-checking and paraphrasing the authors' original content, without suggesting new content. Any content generated with the assistant underwent meticulous manual review and subsequently received final approval from the authors.

## B  COMPRESSION LOSS

**Setup.**  After training an actor $A_\theta$ to *consume* latent communications, we freeze $A_\theta$ and train a reasoning model $M_\phi$ to *produce* compact, information-dense latent communications of length $K$ that the frozen actor can still exploit. For an input instance with supervised token indices $S$ (the teacher-forced window after the first user turn), let

$$H_{1:K}^{\text{gen}} = M_\phi(x) \quad \text{and} \quad H_{1:L}^{\text{full}} = M_{\text{ins}}(x),$$

denote respectively the *generated* latent communication from the trainable reasoning model and the full-length latent communication extracted from a fixed instruction-tuned model $M_{\text{ins}}$. A lightweight communication adapter $g(\cdot)$ (kept frozen) *preprocesses* the latent communication before concatenation with boundary tokens <bop>/<eop>. For brevity, we use $H_K \equiv H_{1:K}^{\text{gen}}$ and $H_L \equiv H_{1:L}^{\text{full}}$.

We define three *actor-scored* forward paths through the frozen actor $A_\theta$ given a prompt $x$: (i) **Path A (generated latents)**: $E^{(A)} = [\, e(x),\, e(\text{<bop>}),\, g(H_K),\, e(\text{<eop>}) \,]$; (ii) **Path D (full-length latents)**: $E^{(D)} = [\, e(x),\, e(\text{<bop>}),\, g(H_L),\, e(\text{<eop>}) \,]$; (iii) **Path B (no latents)**: $E^{(B)} = [\, e(x) \,]$. Let $\ell_t^{(q)}$ be the frozen-actor logits at position $t \in S$ under path $q \in \{A, D, B\}$, and

$$p_t^{(q)} = \text{softmax}\big(\ell_t^{(q)}/T\big)$$

be the corresponding token distributions with temperature $T \geq 1$ used for distillation. Unless stated otherwise, gradients do not flow into $A_\theta$ or $g(\cdot)$. The detailed training procedure is provided in Algorithm 1.

**(1) Actor cross-entropy utility.**  We require the generated message to be *useful* for the frozen actor:

$$\mathcal{L}_{\text{task}} = \frac{1}{|S|} \sum_{t \in S} \big( - \log p_\theta(y_t \mid C_t, H_K) \big) \quad \text{(computed under Path A).}$$

This term enforces that the compressed latents $H_K$ still drive correct next-token predictions, directly penalizing information loss due to shortening ($K \ll L$). It prevents degenerate over-compression that would be efficient but useless to the actor. Practically, it anchors training on task utility so that any compression gain does not come at the cost of downstream performance.

**(2) Uncertainty-weighted agreement.**  We further encourage *behavioral agreement* between using full-length latent communication (Path D) and generated compressed latent communication (Path A), with per-token weights that reflect how much *any* latent reduces uncertainty relative to the no-latent baseline (Path B). Let the entropies be

$$H^{(q)}(t) = - \sum_v p_t^{(q)}(v) \log p_t^{(q)}(v), \quad q \in \{A, D, B\}.$$

Define raw weights $w_t^\star = \max\big(H^{(B)}(t) - H^{(D)}(t),\, 0\big)$ and optionally clip $w_t^\star$ to $[0, \tau]$ to suppress outliers. Normalize to unit mean:

$$w_t = \frac{w_t^\star}{\frac{1}{|S|} \sum_{u \in S} w_u^\star + \varepsilon}.$$

---

[2] https://chat.openai.com/

The agreement term is a temperature-scaled KL:

$$\mathcal{L}_{\text{pref}} \;=\; \frac{1}{\sum_{t \in S} w_t} \sum_{t \in S} w_t \, T^2 \, \text{KL}\!\Big(p_t^{(D)} \,\big\|\, p_t^{(A)}\Big) \;=\; \frac{T^2}{\sum_{t \in S} w_t} \sum_{t \in S} w_t \sum_v p_t^{(D)}(v) \log \frac{p_t^{(D)}(v)}{p_t^{(A)}(v)}.$$

By matching $p^{(A)}$ to $p^{(D)}$ where full latents actually reduce uncertainty (weights $w_t$), this term teaches $H_K$ to reproduce the *informative* behavioral effects of $H_L$ while ignoring positions where latents are unhelpful.

**(3) Latent direction alignment.** To stabilize compression, we align the *global direction* of actor-side latent features induced by generated vs. data latents. Let $Z_k^{(q)} \in \mathbb{R}^{d_z}$ be the actor-side features (after $g(\cdot)$ and the actors input stack) at latent step $k$ under path $q \in \{A, D\}$. When $H_L$ has length $L \neq K$, apply a fixed resampling operator $\rho_K$ (*e.g.,* uniform down/up-sampling) and write $Z_{1:K}^{(D)} = \rho_K\big(Z_{1:L}^{(D)}\big)$. Define step-averaged directions $\bar{z}^{(q)} = \frac{1}{K} \sum_{k=1}^{K} Z_k^{(q)}$ and the cosine penalty

$$\mathcal{L}_{\text{geom}} \;=\; 1 - \cos\!\big(\bar{z}^{(A)}, \, \bar{z}^{(D)}\big) \;=\; 1 - \frac{\langle \bar{z}^{(A)}, \, \bar{z}^{(D)} \rangle}{\|\bar{z}^{(A)}\|_2 \, \|\bar{z}^{(D)}\|_2}.$$

This term preserves the *geometry* of the actor-side representations, preventing the compressed latents from drifting to directions that the actor interprets differently. Empirically, it improves stability and mitigates mode collapse when $K$ is small by retaining the global semantic orientation of $H_L$.

**Overall objective.** The compression objective for $M_\phi$ (with $A_\theta$ frozen) is

$$\mathcal{L}_{\text{compress}} \;=\; \lambda_{\text{task}} \, \mathcal{L}_{\text{task}} \;+\; \lambda_{\text{pref}} \, \mathcal{L}_{\text{pref}} \;+\; \lambda_{\text{geom}} \, \mathcal{L}_{\text{geom}}.$$

In practice, all terms are computed over $t \in S$ with teacher forcing; gradients propagate only to $\phi$.

# C  SET UPS

## C.1  IMPLEMENTATION DETAILS

We implement our method based on the Qwen2.5-7B-Base and Qwen2.5-0.5B-Base models (Yang et al., 2024). All training processes are conducted using mixed-precision training (bfloat16), FlashAttention-2 (Dao, 2023), gradient checkpointing, and DeepSpeed ZeRO-2 (Rajbhandari et al., 2020) with CPU offload for parameters and optimizer states on 8 NVIDIA A100-80G GPUs with batch-level early stopping on the reasoning model. The global batch size is 16, corresponding to 2 samples per GPU. We adopt the AdamW optimizer (Loshchilov & Hutter, 2017) with a learning rate of 1e-5, a 3% warm-up ratio. A validation split of 5% of the dataset is used for model selection. For the pre-process of latent communication, we employ a multi-head attention layer (8 heads), followed by layer normalization and an adaptive projection module. We treat the task loss with unit weight, and scale the other two terms by dynamic coefficients that are annealed during training, with Qwen2.5-7B-Base: $\lambda_{\text{sep}} \in [0.1, 2.0]$, $\lambda_{\text{align}} \in [0.1, 0.2]$ and Qwen2.5-0.5B-Base: $\lambda_{\text{sep}} \in [0.1, 0.5]$, $\lambda_{\text{align}} \in [0.1, 0.2]$. For negative sampling in the contrastive objective, we use latent communication from different tasks within the same batch, which provides a more challenging discrimination target compared to random noise.

For compression training, the weights of three losses are all 1. We implement our system with a frozen actor agent and a trainable reasoning agent based on Qwen2.5-7B-Base. Unless otherwise stated, the student is Qwen2.5-7B and the teacher All runs use mixed precision (bfloat16), FlashAttention-2 (Dao, 2023) when available, gradient checkpointing, and DeepSpeed ZeRO-2 (Rajbhandari et al., 2020) with CPU offload for parameters and optimizer states. We train on $64 \times$A100-80GB GPUs with per-GPU micro-batch size 2; DeepSpeed automatically sets gradient accumulation to meet the target effective batch. The optimizer is AdamW with learning rate $5 \times 10^{-5}$, warmup ratio 3%, and we select models by a 5% validation split with early stopping

We choose Alfworld (Shridhar et al., 2020) to test our Interlat, which provides multi-step tasks that require a multi-agent system to plan and act based on the environment. We adopt the official split with 3119 training tasks, 140 validation tasks, and 134 test tasks. Episodes are capped at 20 environment steps; Success is 1 if the goal state is reached within the budget and 0 otherwise. All models are

trained on ALFWorld trajectory data from (Song et al., 2024), which includes task descriptions, step-by-step thoughts, and actions. Latent communications are extracted from a Qwen2.5-7B-instruct and Qwen2.5-0.5B-instruct model. On our setup, the training time for the actor model is about 6 hours for 7B and 3.3 hours for 0.5B, 8 to 48 hours for the reasoning model for generating length from 8 to 128. We report the mean result over three independent runs

Unless specified, we report the mean result over three independent runs for each model–method/variation pair.

### C.2 BASELINES AND SETTINGS IN INTERLAT.

We consider the following baselines: (1) **CoT (full).** We use complete Chain-of-Thought (CoT) traces produced by a related instruction-tuned model (Qwen2.5-7B-Instruct and Qwen2.5-0.5B-Instruct) to perform full-parameter supervised fine-tuning. In inference, the model receives a complete CoT plan before generating answers. (2) **No-CoT.** The language model is trained to produce the final answer directly, without receiving any plan from other agents.

We also evaluate some variants of our method: (1) **Text.** Instead of latent communication, we feed the corresponding CoT plan (in language space) to the actor. (2) **No-Comm.** We remove any communication from the actors input. (3) **CrossTask.** We replace the current tasks latent communication with one sampled from a different task. (4) **Noised.** We add perturbations to the latent communication $H$: (a) **CovNoise-0.5×/1.0×**: covariance-shaped noise $\varepsilon_t \sim \mathcal{N}(0, \hat{\Sigma})$ with optional strength $\lambda \in \{0.5, 1.0\}$, where $\hat{\Sigma}$ is the sample covariance of the original $H$; (b) **WhiteNoise**: a control drawn from $\mathcal{N}(0, I)$ with the same length. (5) **CovGauss.** We replace the entire $H$ with i.i.d. samples $H_t \sim \mathcal{N}(0, \hat{\Sigma})$ $(0\mu)$ and report a robustness check with $\mathcal{N}(\hat{\mu}, \hat{\Sigma})$ $(\mu)$. These preserve first-second order moments while removing higher-order structure and temporal alignment. (6) **RandomRot.** We apply a structure-preserving but information-scrambling transform $H' = \hat{\mu} + (H - \hat{\mu}) \hat{\Sigma}^{-1/2} Q \hat{\Sigma}^{1/2}$, where $Q$ is a Haar-random orthogonal matrix (Mezzadri, 2006). This preserves the mean/covariance exactly while disrupting higher-order structure.

## D BENCHMARK

Alfworld is a text-only benchmark that simulates embodied household tasks while keeping interaction purely in natural language. Agents observe textual descriptions of the scene and issue high-level commands from a constrained action set (*e.g.,* go to, open, close, take, put, toggle on/off, heat, cool, examine). Tasks are long-horizon and compositional, requiring perception, planning, and execution over multiple steps under partial observability. The benchmark provides official train/validation/test splits and a standard success metric under a fixed step budget (*e.g.,* 20 steps in our setup), enabling systematic and reproducible evaluation of sequential decision-making.

**Task Setup and Evaluation Metrics.** ALFWorld (Shridhar et al., 2020) is a text-based embodied reasoning benchmark where an agent must interact with a simulated household environment to complete goal-oriented tasks (e.g., put the apple in the fridge). Each episode begins with a textual scene description and allows up to 20 environment steps. At every step, the agent observes the updated environment state and issues a textual command from a constrained action set (*go to, open, close, take, put, toggle on/off, heat, cool, examine*, etc.), receiving a textual observation and reward signal. We train agents on trajectories derived from expert demonstrations (Song et al., 2024), which include the environment descriptions, intermediate thoughts, and executed actions. During training, the reasoning agent predicts the next action (or plan) conditioned on task context or received latent communication, while the actor model executes and provides cross-entropy feedback. Alfworld evaluate agents' performance using two primary metrics: success rate and steps. Success rate measures the proportion of tasks in which the final goal state is reached within the allowed step budget (Success = 1 if goal achieved, else 0). Steps reports the average number of environment interactionsi.e., actionobservation cyclestaken to successfully complete or terminate a task, not rounds of inter-agent communication. Higher success rates indicate better reasoning and coordination efficiency. We provide training templates in Appendix J.

**Why ALFWorld for this work?** First, its multi-step, plan-then-act structure closely matches our sender-receiver setup and stresses the precise abilities our method targets: exploration quality, plan following, and coordination. **Prior work shows that continuous latent reasoning is especially advantageous on planning-heavy tasks**, latent representations preserve multiple candidate reasoning branches and promote breadth-first search (BFS) dynamics (Hao et al., 2024). **We ask whether the same or similar advantages hold when agents communicate in latent space;** ALFWorld is an ideal testbed because its tasks require long-horizon planning, where agents iteratively observe, form thoughts, and act based on environment feedback. Second, the text-only interface isolates the communication modality itself, letting us cleanly contrast language space vs. latent space communication without confounds from external tools or perception pipelines, thereby allowing us to probe what new properties latent communication introduces for agent behavior. Third, community resources provide consistent task descriptions and action trajectories, allowing us to derive both language baselines (*e.g.,* a CoT plan) and latent representations from the same underlying data, reducing distribution shift. Finally, the moderate episode length and standardized protocol make it feasible to average over multiple independent runs, yielding robust statistics for ablations and compression analyses.

## E  ABLATIONS AND STEP ANALYSIS

We present ablation studies for both the actor and reasoning models, reporting average steps for successful trials versus all trials (success/all). As a complement to our main ablation results, this section analyzes the step count to provide deeper insights into agent behavior.

As shown in Table 4, the results reveal a nuanced relationship between step count and performance. On seen tasks, ablating components results in a lower overall success rate. Interestingly, although these models take fewer steps on the trials they do complete, their high failure rate indicates an inability to properly interpret the latent communication and reliably solve tasks. These results support our findings that information-rich latent communication encourages more effective and thorough exploration.

On unseen tasks, several ablations (*e.g.,* removing the curriculum or the adapter) exhibit an opposite pattern: the agent takes more steps yet achieves a lower success rate. This demonstrates that longer trajectories do not necessarily equate to productive exploration. Without these crucial components, the agents policy tends to wander without forming effective task-solving strategies. Therefore, the additional steps reflect an unstructured, inefficient search rather than the deliberate exploration enabled by our method. This analysis underscores the importance of evaluating step count in conjunction with the success rate for a holistic assessment of an agent's true performance.

| Method | Seen | Steps | Unseen | Steps |
|---|---|---|---|---|
| **Actor model** | | | | |
| Ours Full | $\mathbf{70.48}_{\pm 1.01}$ | 9.41/12.54 | $\mathbf{65.42}_{\pm 0.87}$ | 9.86/13.37 |
| w/o curri | $33.10_{\pm 2.97}$ | 9.07/16.38 | $20.65_{\pm 2.15}$ | 10.47/18.03 |
| w/o $\mathcal{L}_{\text{sep}}$ | $\underline{58.81}_{\pm 1.41}$ | 8.07/12.98 | $\underline{60.70}_{\pm 5.50}$ | 9.64/13.71 |
| w/o $\mathcal{L}_{\text{align}}$ | $56.90_{\pm 1.41}$ | 8.16/13.26 | $53.98_{\pm 3.35}$ | 9.56/14.36 |
| w/o adapter | $4.05_{\pm 1.70}$ | 9.32/19.57 | $4.48_{\pm 1.31}$ | 10.53/19.58 |
| **Reasoning model** | | | | |
| Ours Full | $\mathbf{68.10}_{\pm 1.93}$ | 9.21/12.65 | $\underline{62.94}_{\pm 2.03}$ | 9.88/13.63 |
| w/o $\mathcal{L}_{\text{task}}$ | $\underline{65.71}_{\pm 1.43}$ | 8.86/12.68 | $\mathbf{63.18}_{\pm 3.47}$ | 9.68/13.48 |
| w/o $\mathcal{L}_{\text{pref}}$ | $64.76_{\pm 2.97}$ | 8.92/12.82 | $60.20_{\pm 3.13}$ | 9.68/13.79 |
| w/o $\mathcal{L}_{\text{geom}}$ | $64.05_{\pm 3.55}$ | 8.71/12.77 | $59.45_{\pm 3.01}$ | 9.88/13.98 |

Table 4: Ablation of training components. Ours Full uses all components.

---

**Algorithm 1** Two-Stage Training for Latent Communication

---

**Require:** Dataset $\mathcal{D}$; actor $A_\theta$; teacher $M_{\text{ins}}$; boundary tokens <bop>, <eop>;
**Require:** replacement schedule $r_t$;
**Require:** loss weights $(\lambda_{\text{CE}}, \lambda_{\text{align}}, \lambda_{\text{sep}})$

**Stage I: Teach the actor to consume latents (no compression)**

1: **for** epoch $= 1 \rightarrow E_1$ **do**
2:     **for** $(x, y) \sim \mathcal{D}$ **do**
3:         $(H, P) \leftarrow M_{\text{ins}}(x)$            $\triangleright$ data latents and plan from the frozen teacher
4:         $H^{(r)} \leftarrow \text{Curriculum}(H, r_t)$          $\triangleright$ random curriculum; length preserved
5:         $E \leftarrow [\, e(x), e(\text{<bop>}), g(H^{(r)}), e(\text{<eop>}) \,]$          $\triangleright$ latent-conditioned input
6:         $E^{(\text{plan})} \leftarrow [\, e(x), e(\text{<bop>}), e(P), e(\text{<eop>}) \,]$          $\triangleright$ plan-only input
7:         $S \leftarrow \text{SupervisedPositions}(y)$          $\triangleright$ token indices within the supervised window
8:         $(\ell_A^{(r)}[S], p_A^{(r)}[S]) \leftarrow \text{Forward}(A_\theta, E, S)$          $\triangleright$ teacher forcing
9:         $(\ell_{\text{plan}}[S], p_{\text{plan}}[S]) \leftarrow \text{Forward}(A_\theta, E^{(\text{plan})}, S)$
10:       sample $\tilde{H}$ from another task/batch;
11:       $\tilde{E} \leftarrow [\, e(x), e(\text{<bop>}), g(\tilde{H}), e(\text{<eop>}) \,]$
12:       $(\ell_A^{(\text{neg})}[S], p_A^{(\text{neg})}[S]) \leftarrow \text{Forward}(A_\theta, \tilde{E}, S)$
13:       $\mathcal{L}_{\text{total}}^{\text{Stage I}} \leftarrow \text{TotalLoss}_{\text{Stage I}}\Big( p_A^{(r)}[S], y[S], p_{\text{plan}}[S],$

$$\ell_A^{(r)}[S], \ell_{\text{plan}}[S], p_A^{(\text{neg})}[S]; \alpha, \beta, \lambda_{\text{CE}}, \lambda_{\text{align}}, \lambda_{\text{sep}} \Big)$$

14:       $\theta \leftarrow \theta - \eta \nabla_\theta \mathcal{L}_{\text{total}}^{\text{Stage I}}$
15:     **end for**
16: **end for**

**Stage II: Train the reasoner to compress (freeze $A_\theta$)**

17: **for** epoch $= 1 \rightarrow E_2$ **do**
18:     **for** $(x, y) \sim \mathcal{D}$ **do**
19:         $H_K \leftarrow M_\phi(x)$
20:         $H_L \leftarrow \text{stopgrad}(M_{\text{ins}}(x))$
21:         $E^{(A)} \leftarrow [\, e(x), e(\text{<bop>}), g(H_K), e(\text{<eop>}) \,]$
22:         $E^{(D)} \leftarrow [\, e(x), e(\text{<bop>}), g(H_L), e(\text{<eop>}) \,]$
23:         $(\ell^{(A)}[S], p^{(A)}[S]) \leftarrow \text{Forward}(A_\theta, E^{(A)}, S)$          $\triangleright$ no grad into $A_\theta$
24:         $(\ell^{(D)}[S], p^{(D)}[S]) \leftarrow \text{Forward}(A_\theta, E^{(D)}, S)$          $\triangleright$ stop-grad
25:         $(\ell^{(\text{base})}[S], p^{(\text{base})}[S]) \leftarrow \text{Forward}(A_\theta, [\, e(x) \,], S)$ $\triangleright$ context-only baseline; stop-grad
26:         $\mathcal{L}_{\text{total}}^{\text{Stage II}} \leftarrow \text{TotalLoss}\Big( p^{(A)}[S], y[S], p^{(D)}[S], p^{(\text{base})}[S], w[S]; \lambda_{\text{CE}}, \lambda_{\text{pref}}, \tau \Big)$
27:         $\phi \leftarrow \phi - \eta \nabla_\phi \mathcal{L}_{\text{total}}^{\text{Stage II}}$
28:     **end for**
29: **end for**

---

**Algorithm 2** Inference with Latent Communication (training-free or trained)

---

**Require:** Dataset $\mathcal{D}$; input $x$; reasoner $M_\phi$; actor $A_\theta$; boundary tokens; target length $K$

1: $H_K \leftarrow M_\phi(x)$
2: $E \leftarrow [\, e(x), e(\text{<bop>}), g(H_K), e(\text{<eop>}) \,]$
3: $\hat{y} \leftarrow \text{Decode}(A_\theta, E)$
4: **return** $\hat{y}$

---

## F  TRAINING AND INFERENCE PSEUDOCODE

Algorithm 1 gives the detailed two-stage first actor agent, second reasoning agent training procedure for latent communication, and Algorithm 2 summarizes the inference process.

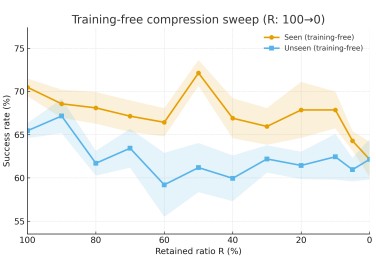

(a) Training-free sweep over retained ratio $R$

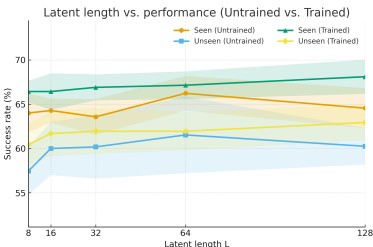

(b) Latent length $L$ vs. performance (untrained & trained)

Figure 8: Result of compression on seen and unseen tasks. **Left:** Success rate under training-free compression with different retained ratios $R$. **Right:** Performance of untrained and trained models across latent lengths $L$

| Ratio | Seen | Steps | Unseen | Steps | Time |
|-------|------|-------|--------|-------|------|
| Untrained | | | | | |
| Full | $70.48_{\pm 1.01}$ | 9.41/12.54 | $65.42_{\pm 0.87}$ | 9.86/13.37 | 9.19s |
| 90% | $68.57_{\pm 1.63}$ | 8.77/12.30 | $\mathbf{67.16}_{\pm 1.97}$ | 9.27/12.79 | - |
| 80% | $68.10_{\pm 1.83}$ | 8.56/12.21 | $61.69_{\pm 1.43}$ | 9.10/13.28 | - |
| 70% | $67.14_{\pm 1.82}$ | 8.68/12.40 | $63.43_{\pm 2.24}$ | 9.42/13.29 | - |
| 60% | $66.43_{\pm 1.63}$ | 8.52/12.37 | $59.20_{\pm 3.69}$ | 9.90/14.02 | - |
| 50% | $\mathbf{72.14}_{\pm 1.48}$ | 9.03/12.09 | $61.19_{\pm 2.84}$ | 9.37/13.50 | - |
| 40% | $66.90_{\pm 2.31}$ | 8.88/12.56 | $59.95_{\pm 2.64}$ | 9.52/13.72 | - |
| 30% | $65.95_{\pm 2.12}$ | 8.80/12.61 | $62.19_{\pm 1.58}$ | 10.11/13.85 | - |
| 20% | $67.86_{\pm 3.23}$ | 8.97/12.52 | $61.44_{\pm 1.58}$ | 9.98/13.84 | - |
| 10% | $67.86_{\pm 2.12}$ | 8.76/12.37 | $62.44_{\pm 2.64}$ | 9.72/13.58 | - |
| 5% | $64.52_{\pm 1.12}$ | 9.19/13.02 | $60.95_{\pm 1.35}$ | 9.90/13.84 | - |
| 0% | $62.14_{\pm 2.01}$ | 10.19/13.90 | $62.19_{\pm 2.32}$ | 10.23/13.92 | - |
| 128L | $64.52_{\pm 2.26}$ | 8.68/12.70 | $60.20_{\pm 2.06}$ | 9.69/13.79 | 3.55s |
| 64L | $66.19_{\pm 1.95}$ | 8.76/12.56 | $61.44_{\pm 4.32}$ | 9.85/13.76 | 1.83s |
| 32L | $63.57_{\pm 2.01}$ | 8.66/12.79 | $60.20_{\pm 3.58}$ | 9.87/13.90 | 1.03s |
| 16L | $64.29_{\pm 1.34}$ | 8.64/12.70 | $59.95_{\pm 3.01}$ | 10.07/14.05 | 0.62s |
| 8L | $64.05_{\pm 2.18}$ | 8.80/12.83 | $57.46_{\pm 2.69}$ | 10.29/14.42 | 0.39s |
| Trained | | | | | |
| 128L | $68.10_{\pm 1.93}$ | 9.21/12.65 | $62.94_{\pm 2.03}$ | 9.88/13.63 | 2.25s |
| 64L | $67.14_{\pm 1.56}$ | 9.15/12.72 | $61.94_{\pm 2.13}$ | 9.92/13.76 | 1.16s |
| 32L | $66.90_{\pm 1.46}$ | 9.02/12.65 | $61.94_{\pm 2.56}$ | 9.96/13.78 | 0.60s |
| 16L | $66.43_{\pm 2.05}$ | 9.08/12.75 | $61.69_{\pm 2.56}$ | 9.98/13.82 | 0.33s |
| 8L | $66.43_{\pm 1.22}$ | 9.11/12.77 | $60.45_{\pm 2.23}$ | 9.90/13.89 | 0.20s |

Table 5: Complete compression results with seen/unseen accuracy, steps, and latency across tasks.

## G COMPRESSION RESULT

In this section, we provide more detailed results on compression with average steps as success/all across tasks in Table 5 and corresponding performance trend in Figure 8. Latency is measured on the same machine and decoding policy (if needed) across rows [3].

---

[3] For the untrained reasoning model, we use the standard `generate` API from Hugging Face `transformers`; see https://github.com/huggingface/transformers.

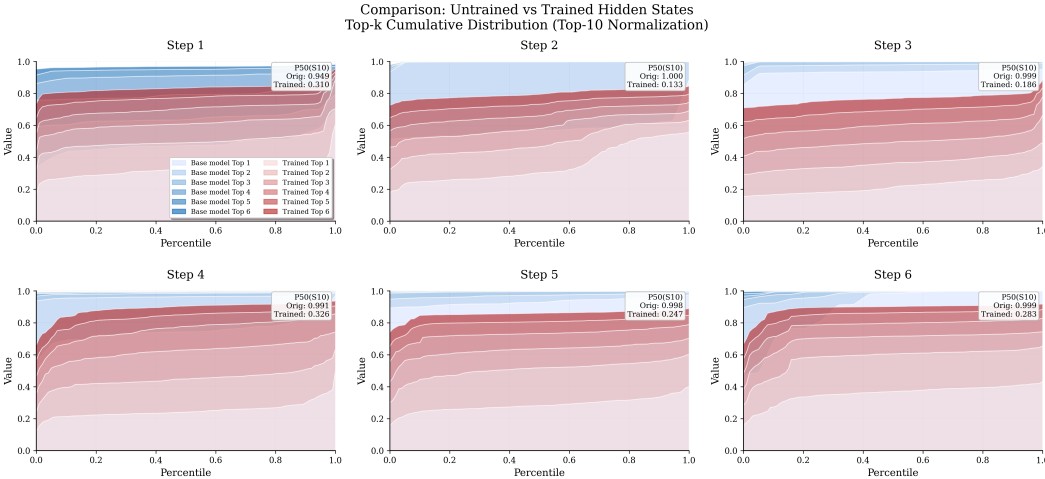

Figure 9: Parallelism in latent communication over the first six steps. Red indicates latents from the trained model, and blue indicates latents from the untrained base model. The trained latents preserve stable vertical gaps between successive Top-$k$ bands and achieve a markedly lower $P_{50}(S_{10})$, evidencing persistent parallelism, whereas the untrained base model's latents progressively collapse toward Top-1.

## H    LATENT PARALLELISM ANALYSIS

We first compared the latent communications produced by our trained reasoning model with those from an off-the-shelf Qwen2.5-7B-Instruct model in the compression-effectiveness analysis (see the Experiments section). Because our reasoning model is initialized from Qwen2.5-7B-Base, we additionally compare it with this base model, which has not been trained for generating compressed latent communication, in Figure 9. The findings are consistent with the earlier comparison: the trained model maintains stable vertical gaps between successive Top-$k$ curves across steps and exhibits a substantially lower $P_{50}(S_{10})$, whereas the base model shows a clear convergence toward Top-1.

We further extend the parallelism analysis to a deeper horizon of 32 steps. As shown in Figure 10, the trained model exhibits stable vertical gaps between successive Top-$k$ curves throughout these steps. This extended analysis further verifies that the trained latent representations preserve a broader set of plausible reasoning paths by sustaining a more balanced probability distribution rather than prematurely collapsing to a Top-1 hypothesis.

## I    QUALITATIVE ANALYSIS OF LATENT COMMUNICATION VIA T-SNE

To provide a qualitative understanding of the semantic structure encoded within our latent communications, we performed a t-SNE visualization on 3,119 samples from the ALFWorld training set. Each sample corresponds to the mean-pooled last-layer hidden states generated by the reasoning agent for a specific task.

The tasks are categorized according to the official ALFWorld task templates, which define six core reasoning patterns: 'pick_and_place', 'pick_clean_then_place', 'pick_heat_then_place', 'pick_cool_then_place', 'look_in_recep', and 'look_at_obj'. As shown in the accompanying figure, the t-SNE plot reveals distinct, albeit overlapping, clusters corresponding to these different task templates. For instance, the 'pick_and_place' cluster (teal) occupies a central region, while 'pick_heat_then_place' (pink) and 'pick_cool_then_place' (blue) form separate, adjacent clusters, suggesting that the model distinguishes between actions requiring thermal manipulation. Similarly, the 'look_in_recep' (orange) and 'look_at_obj' (red) clusters, though smaller due to lower frequency, also exhibit localized concentrations.

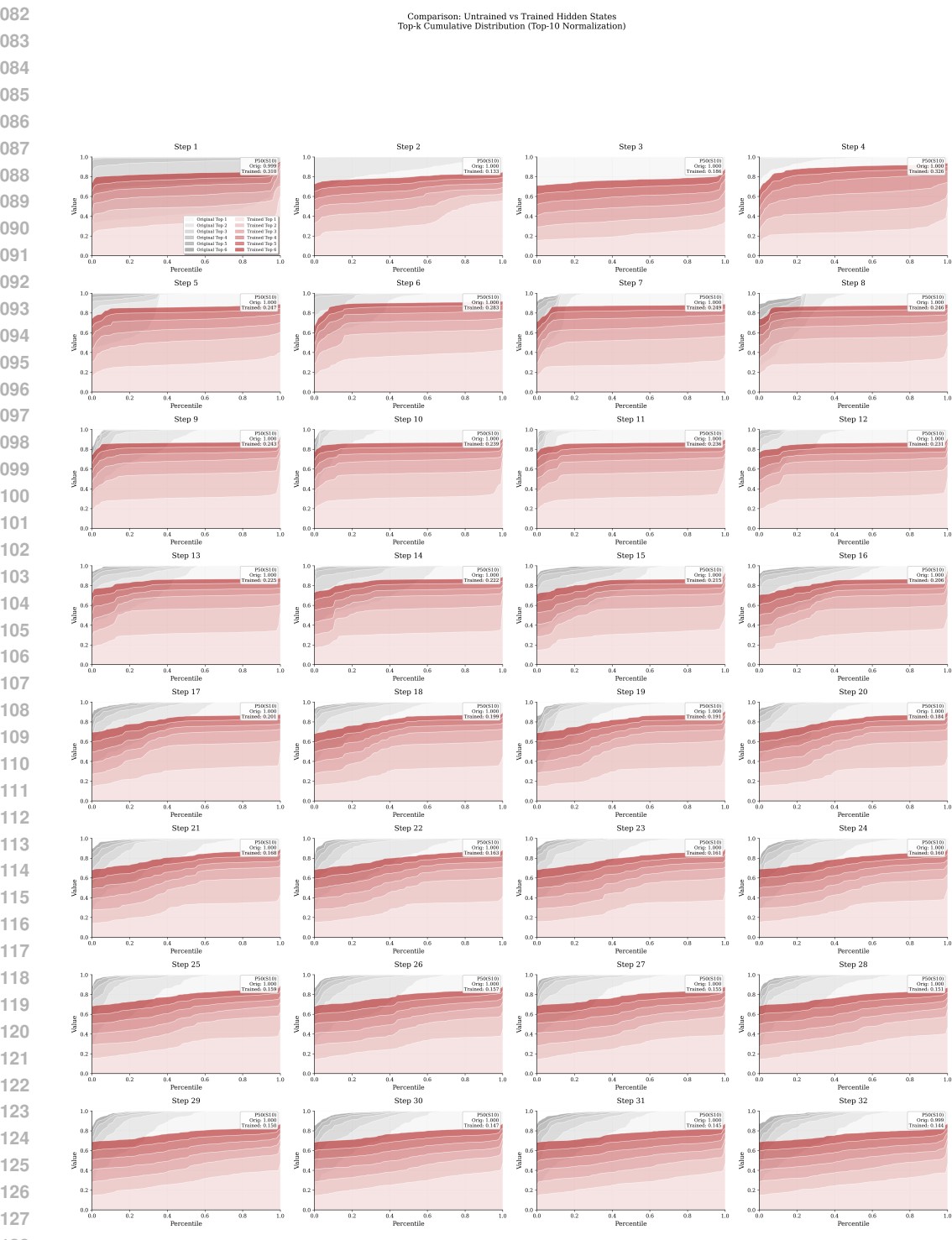

Figure 10: **Extended analysis (32 steps).** Same construction as Fig. 4, now for steps 1–32. Persistent separation among successive Top-$k$ bands and consistently lower $\text{P50}(S_{10})$ values indicate that the trained latents maintain broad, plausible reasoning branches across the entire sequence, despite compression.

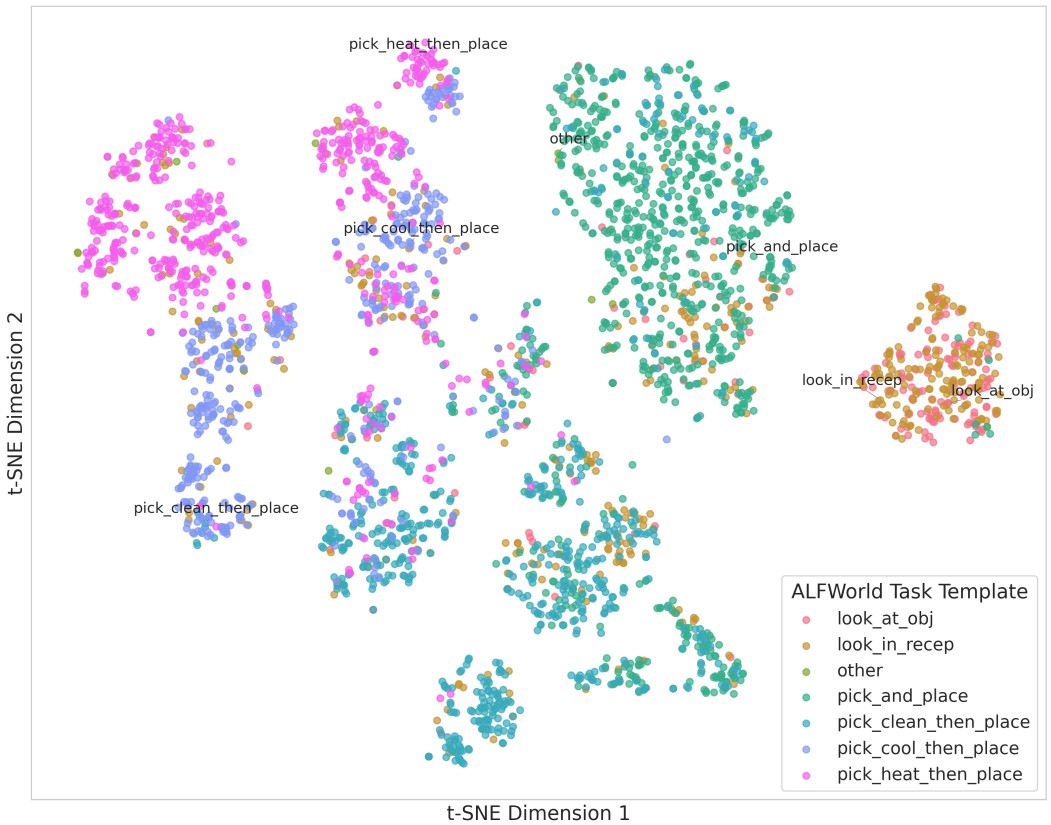

Figure 11: t-SNE visualization of latent communications grouped by ALFWorld task template (N=3,119). Each point represents the mean-pooled last-layer hidden state of a reasoning agents plan for a specific task. Colors and labels correspond to six core task templates: pick_and_place, pick_clean_then_place, pick_heat_then_place, pick_cool_then_place, look_in_recep, and look_at_obj. The distinct clustering patterns demonstrate that Interlats latent communication encodes rich, task-specific semantic information, enabling the actor agent to differentiate between diverse reasoning paradigms without relying on natural language.

As shown in Figure 11. This clustering pattern indicates that latent communications are not random noise but encode rich, task-specific semantics that can be effectively interpreted by the actor model. The spatial separation between clusters corresponds to differences in procedural complexitysuch as simple placement versus multi-step actions involving cleaning, heating, or searchingcaptured solely through latent representations without explicit language. Moreover, the intra-cluster dispersion (*e.g.*, within pick_and_place) illustrates that even within a single task type, latent vectors preserve nuanced variations across objectreceptacle combinations, demonstrating that latent communication maintains diverse reasoning paths rather than collapsing to a single representation.

In summary, this qualitative analysis confirms that Interlat s latent communication mechanism successfully transmits meaningful, structured information about the reasoning models internal reasoning process, enabling the receiver to perform sophisticated, task-appropriate behaviors without relying on explicit natural language.

## J  TRAINING TEMPLATE

We present an example in Figure J to illustrate how agents explore and solve tasks in Alfworld. After perceiving the environment, the agent executes an action, receives feedback from the environment, and then proceeds to the next step until the goal is accomplished. Figure J shows an example of how training data is structured for the actor agent. We append either the latent communication or the natural-language plan after the instruction to facilitate inter-agent communication.

**Conversations**

**Env**
You are in the middle of a room. Looking quickly around you, you see a bathtubbasin 1, a cabinet 2, a cabinet 1, a countertop 1, a garbagecan 1, a handtowelholder 1, a sinkbasin 1, a toilet 1, a toiletpaperhanger 1, and a towelholder 1.
**Task:** put a toiletpaper in toiletpaperhanger.

**Agent**
**Thought**: I need to place a toilet paper roll on the toilet paper hanger. First, I should check if there's already a roll on the hanger or if it's empty.
**Action**: go to toiletpaperhanger 1

**Env**
**Observation**: On the toiletpaperhanger 1, you see nothing.

**Agent**
**Thought**: The toilet paper hanger is empty, so I need to find a new roll of toilet paper to put on it.
**Action**: go to toilet 1

**Env**
**Observation**: On the toilet 1, you see a soapbottle 1, and a toiletpaper 1.

**Agent**
**Thought**: I've found a toilet paper roll on the toilet. I should pick it up so I can place it on the hanger.
**Action**: take toiletpaper 1 from toilet 1

**Env**
**Observation**: You pick up the toiletpaper 1 from the toilet 1.

**Agent**
**Thought**: Now that I have the toilet paper roll, I need to return to the toilet paper hanger to install it.
**Action**: go to toiletpaperhanger 1

**Env**
**Observation**: On the toiletpaperhanger 1, you see nothing.

**Agent**
**Thought**: I'm back at the toilet paper hanger with the roll in hand. It's time to put the toilet paper on the hanger.
**Action**: put toiletpaper 1 in/on toiletpaperhanger 1

---

**Training Template for Actor Agent**

```
<im_start>system
```
You are a helpful assistant. `<im_end>`
`<im_start>user`
Interact with a household to solve a task. Imagine you are an intelligent agent in a household environment and your target is to perform actions to complete the task goal. At the beginning of your interactions, you will be given a detailed description of the current environment and your goal to accomplish.
For each of your turns, you will be given the observation of the last turn. You should first think about the current condition and plan for your future actions, and then output your action in this turn. Your output must strictly follow this format:"Thought: your thoughts.
Action: your next action".
The available actions are:
1. go to recep
2. task obj from recep
3. put obj in/on recep
4. open recep
5. close recep
6. toggle obj recep
7. clean obj with recep
8. heat obj with recep
9. cool obj with recep
where obj and recep correspond to objects and receptacles.
After your each turn, the environment will give you immediate feedback based on which you plan your next few steps. if the envrionment output "Nothing happened", that means the previous action is invalid and you should try more options.
Your response should use the following format:
Thought: <your thoughts>
Action: <your next action>
You are in the middle of a room. Looking quickly around you, you see a armchair 2, a armchair 1, a coffeetable 2, a coffeetable 1, a diningtable 1, a garbagecan 1, a sidetable 2, a sidetable 1, and a sofa 1.
Your task is to: find two newspaper and put them in armchair.
Now, you are given a step-by-step plan to complete this task as follow:
<bop>
Insert a latent communication or textual plan for inter-agent communication.

```
<eop>
<im_end>
<im_start>assistant
Thought: To complete the task, I need to locate two newspapers and place them in an arm-
chair. I should start by searching the nearby furniture for any newspapers.
Action: go to sofa 1 <im_end>
<im_start>user
Observation: On the sofa 1, you see a box 3, a box 2, a newspaper 2, and a pillow 1. <im_end>
<im_start>assistant
Thought: I've found one newspaper on sofa 1. I should pick it up to place it in the armchair
later.
Action: take newspaper 2 from sofa 1 <im_end>
<im_start>user
Observation: You pick up the newspaper 2 from the sofa 1. <im_end>
<im_start>assistant
Thought: With newspaper 2 in hand, I need to find the second newspaper. The dining table is
another common place where newspapers might be found.
Action: go to dining table 1 <im_end>
<im_start>user
Observation: On the dining table 1, you see a box 1, a credit card 2, a credit card 1, a laptop
1, a newspaper 1, a plate 3, a plate 1, a remote control 1, a statue 3, a statue 1, a tissue box 3,
and a tissue box 1. <im_end>
<im_start>assistant
There's the second newspaper on the dining table. I should take it so I can place both news-
papers in the armchair as instructed.
Action: take newspaper 1 from dining table 1 <im_end>
<im_start>user
Observation: You pick up the newspaper 1 from the dining table 1. <im_end>
<im_start>assistant
Thought: With both newspapers now in my possession, I need to find an armchair to place
them in.
Action: go to armchair 1 <im_end>
<im_start>user
Observation: On the armchair 1, you see a keychain 1. <im_end>
<im_start>assistant
Thought: The armchair has a keychain on it, but there's enough space for the newspapers. I'll
place the first newspaper here.
Action: put newspaper 2 in/on armchair 1 <im_end>
<im_start>user
Observation: You put the newspaper 2 in/on the armchair 1. <im_end>
<im_start>assistant
Thought: With the first newspaper placed, I need to put the second newspaper in the armchair
to complete the task.
Action: put newspaper 1 in/on armchair 1 <im_end>
```

