# OpenReview forum: "Enabling Agents to Communicate Entirely in Latent Space"
_ICLR.cc/2026/Conference — ICLR 2026 Conference Withdrawn Submission_

### Official Review · Reviewer_1RDo · 2025-10-27

**Soundness:** 3
**Presentation:** 1
**Contribution:** 3
**Rating:** 4
**Confidence:** 2

**Summary:**

This paper proposes to train LLM agents in a collaborative framework in which a sender communicates in the latent space and an actor completes the task based on received messages. The authors combine multiple learning objectives, curriculum learning, and information compression in the framework to train the reasoning model of the sender with frozen-actor supervision from the task environment. Evaluation results show that agents trained to communicate in latent space outperform baselines where agents are supervised fine-tuned with CoT or direct natural language communication. Ablation experiments were conducted to evaluate the impact of information compression and other modules in the proposed framework.

**Strengths:**

1. The general idea of allowing LLM agents to communicate in a latent space instead of the constrained natural language space is novel and intuitive.

2. The proposed method outperforms the baseline and ablated conditions in the evaluation benchmark.

**Weaknesses:**

1. The framework of training agents to communicate in a high-dimensional space to solve the sender-receiver task has been extensively researched in the emergent communication and multi-agent reinforcement learning (MARL) community. I would recommend the authors include a thorough literature review and position this work among previous work, highlighting its contributions beyond simply replacing the policy model with a transformer-based LLM. Attached are a few representative papers to start with:

a. Multi-agent cooperation and the emergence of (natural) language

b. Emergence of linguistic communication from referential games with symbolic and pixel input

c. Emergent discrete communication in semantic spaces

d. Trading off Utility, Informativeness, and Complexity in Emergent Communication

2. The methods section is hard to follow. I would recommend providing a more specific description of the proposed framework for readers with a more general background.

a. The description of the training framework could be more specific for better reproducibility. For example, how is the permutation done in conditional mind separation? What is the language space plan in plan-aligned regulation? How does stochastic replacement happen in curriculum learning?

b. The motivation and description of information compression are too brief to be comprehensible. It is not clear how the compression is done in Section 3.3 of the main text. Even after reading Appendix B, the use of a fixed instruction-tuned model and the distillation process lacks a clear rationale. I would recommend moving the description of compression to the main text and emphasizing the motivation for this module.

c. It is not clear how the environment reward backpropagates through the sender-receiver setup. Is reinforcement learning used to train the agent in an interactive environment, or is this offline training with a fixed trajectory dataset?

d. In the main text, the actor is described as frozen during training. However, the pseudocode in Appendix F shows a two-stage training framework. Please clarify this discrepancy.

e. The "training-free" and "trained" settings are not explained before their use in Section 4.2.

3. Minor format issue: The font size in Figure 1 should be enlarged for better readability.

**Questions:**

1. How were the MHA and Projector trained?

2. There seems to be an assumption that the fixed actor can understand the latent communication of the reasoning model because they have the same backbone (e.g., Qwen2.5). Would the findings reported in the paper still hold if these two modules were based on different models?

---

> ### Author Response · Authors · 2025-11-15
>
> Thank you for your thorough review and feedback. Below is a detailed point-by-point response addressing your main concerns. **Please refer to the newly uploaded revised version of the paper to locate the corresponding line numbers.**
>
> ---
>
> First of all, we want to make some explaination around your concerns raised in weaknesses section.
>
> ---
>
> Weakness 1:
>
> While our formulation indeed follows the sender–receiver paradigm, **Interlat differs fundamentally from these prior studies in both learning objective and representation form**.
>
> Classical emergent-communication research—such as *multi-agent cooperation and the emergence of language*, *referential games with symbolic or pixel input*, *emergent discrete communication in semantic spaces*, and *utility–informativeness trade-offs*—focuses on **learning discrete symbolic protocols** through reinforcement signals or differentiable relaxation. These works aim to evolve *communication languages* that agents invent from scratch to maximize shared reward or efficiency within simplified simulated environments.
>
> Unlike prior emergent-communication studies that learn **symbolic or discrete messages** for coordination, **Interlat enables direct reasoning pathways and information transfer between agents** via their hidden states. As described in **Section 3.1 (lines 162–215)**, the sender model produces a sequence of last-layer hidden states (H=[h_1,\dots,h_L]) that encode **intermediate reasoning steps**, not modality embeddings or symbolic tokens. The receiver then conditions on these latent reasoning trajectories to generate actions. This differs fundamentally from **representation learning**, where the goal is to embed external sensory inputs (e.g., images) into static feature vectors. In contrast, our latent vectors represent **dynamically generated cognitive states** that carry causal reasoning information.
>
> Furthermore, as outlined in related work, we position Interlat beyond previous works that graft activations or exchange state deltas (Ramesh & Li 2025; Tang et al. 2025) by introducing a structured, fully latent-space channel with compression and alignment objectives. These contributions extend the sender–receiver paradigm to the **latent reasoning domain** of large models, enabling efficient, language-free inter-agent collaboration.

---

> ### Author Response · Authors · 2025-11-15
>
> ### **Weakness 2: Response to Reviewer Concern: Clarity and Reproducibility of the Methods Section**
>
> We thank the reviewer for the detailed feedback. We clarify each point below and emphasize that all mentioned components are explicitly defined and reproducible in the current submission.
>
> ---
>
> **(a) Specificity of the training framework**
>
> - **Conditional mind separation (CMS):**
>
>     As described in line 177, **the permutation is conducted *within-training-batch* by shuffling the latent states ($H$) among samples that share the same task context ($C_t$), as in the training process, the batch size is more than 1. This creates mismatched pairs ($C_t, \tilde H$) for the Jensen–Shannon separation loss, ensuring the divergence reflects representational specificity rather than task variance.
>
> - **Plan-aligned regulation (PAR):**
>
>     In the first training stage, the reasoning model still generates messages in the **language space**, producing both textual tokens and their corresponding hidden states. As shown in Figure 1 in our paper, we collect these hidden states to form latent communications, while the **language-space plan** refers to the sequence of generated **text tokens** (i.e., the natural-language plan) that corresponds one-to-one with those latent states. It is used as a soft constraint to regularize the receiver’s prediction distribution ($p_\theta(\cdot|C_t, H)$) toward that of ($p_\theta(\cdot|C_t, P)$). This bridges latent- and text-based communication channels during early curriculum stages.
>
> - **Stochastic replacement in curriculum learning:**
>
>     The *stochastic replacement* process is explicitly described in line 197.
>
>     Concretely, during training, we *gradually replace* portions of the latent communication with their corresponding text embeddings in a **left-to-right** fashion. For each training instance, we randomly sample a replacement rate
>
>     ($r \sim p_R$) from a uniform set ($R = {0, 0.1, \dots, 1.0}$), and construct a mixed embedding sequence $H^{(r)} = h_{1:r\times L} \oplus e_{r\times L+1:L}$, where ($h_i$) are latent states and ($e_i$) are text embeddings.
>
>     ---
>
>
> ### **(b) Motivation and mechanism of information compression**
>
>  The motivation and mechanism of **information compression** are detailed in **Section 3.3 “Information Compression” ** and further expanded in **Appendix B**.
>
> Below we clarify both **the motivation** and **how compression is implemented**:
>
> - **Motivation.**
>
>     Latent communication between agents is high-dimensional and information-dense. We aim to *compress* these latent communications without losing essential task-relevant information, thereby reducing communication latency and compute cost.
>
>     As described in lines 86, the latent channel (≈ 40 k bits/state) can encode far richer semantics than discrete tokens (≈ 15 bits/token). Our goal is thus to learn *shorter but information-preserving* latent representations (H_K) that remain interpretable by the actor agent, achieving up to a 24× speed-up while maintaining comparable success rates (Table 2, lines 377 – 377).
>
> - **Compression process.**
>
>     As defined in Eq. (2), we train the reasoning model ($M_\phi$) to *autoregressively feed its own hidden states as next-step embeddings*:
>
>     $\langle M_\phi(E_i) \rightarrow h_i, \quad E_{i+1} = E_i \oplus \mathrm{Proj}(h_i)\rangle_{\circlearrowleft}$,
>
>     where ($\mathrm{Proj}(\cdot)$) is a lightweight LayerNorm + Linear bridge.
>
>     During this stage, the **actor model and communication adapter are frozen**, and only the reasoning model is optimized to generate shorter latent sequences ($H_K$) that still yield correct actions when consumed by the actor.
>
>     The complete loss formulation is provided in Appendix B (lines 648–755). In the revised version, we will move the essential components of this formulation to the main text for better accessibility.
>
> - **Why a fixed instruction-tuned model.**
>
>     The instruction-tuned model provides a **teacher latent communication ($H_L$)** that serves as the full-length reference. Using a frozen, instruction-tuned teacher ensures that the compressed latent ($H_K$) inherits consistent information grounding and avoids drift in meaning across tasks. The reasoning model thus learns to distill *the behavioral effect* of the teacher’s full latent communication into a shorter, efficient sequence, guided by the frozen actor’s supervision (Appendix B).

---

> ### Author Response · Authors · 2025-11-15
>
> ### **(c) Reward backpropagation and training setting**
>
> **Our training is fully offline and supervised**, *not reinforcement learning (RL)*. The environment reward in ALFWorld is **not directly backpropagated** through the agents; instead, it is used only for *evaluation*. The entire learning process relies on **teacher-forced supervision** over fixed task trajectories.
>
> - **Training paradigm:**
>
>     As described in **Training Procedure**and in **Appendix C.1**, both the *reasoning (sender)* and *actor (receiver)* agents are trained via **cross-entropy and contrastive (JS separation / plan alignment)** objectives using **pre-collected ALFWorld trajectories** from **[1]**, we provided training example in Appendix I (lines 1049).
>
>     The task loss ($L_{\text{task}}$) (Eq. (1)) is a *supervised next-token cross-entropy* over these trajectories, ensuring correct language/action prediction given the latent communication (H).
>
>     There is **no policy gradient or environment sampling** during training.
>
> - **Role of environment reward:**
>
>     The *success rate* from ALFWorld—i.e., whether the goal state is achieved within 20 steps—is reported **only at evaluation time** as a performance metric.
>
>     In other words, the environment reward is *not differentiable* and *not backpropagated* through the sender–receiver pipeline.
>
>     The gradients flow solely from supervised objectives ($L_{\text{task}}, L_{\text{sep}}, L_{\text{align}}$) defined in the offline dataset.
>
> - **Why offline training:**
>
>     Our goal in this paper is to **isolate the effects of latent-space communication** rather than confound them with RL optimization.
>
>     Training with fixed trajectories allows us to (1) control for environmental randomness, (2) measure stability and transfer under latent compression, and (3) ensure reproducibility—consistent with our focus on communication efficiency rather than policy learning.
>
>
> **[1]** Yifan Song, Da Yin, Xiang Yue, Jie Huang, Sujian Li, and Bill Yuchen Lin. Trial and error:
> Exploration-based trajectory optimization for llm agents. In ACL 2024.
>
> ---
>
> ### **(d) Frozen actor and two-stage framework**
>
> There is **no contradiction** between the frozen actor and the two-stage procedure.
>
> The *actor being frozen* and the *two-stage framework* are not contradictory, they refer to **different phases** of the overall training pipeline, as clarified below.
>
> - **Stage 1: Actor training (learns to interpret latent communications).**
>
>     In the first stage, both **reasoning (sender)** and **actor (receiver)** models are trained *jointly* on pre-collected trajectories to enable the actor to interpret latent messages.
>
>     This process corresponds to **Section 3.2**, where the actor’s total loss combines:
>
>     $\mathcal{L}_{\text{total}} = \mathcal{L}_{\text{task}} + \lambda_S \mathcal{L}_{\text{sep}} + \lambda_A \mathcal{L}_{\text{align}}$.
>
>     Here, gradients flow through the actor network, allowing it to *learn how to decode and utilize latent states*.
>
>     This corresponds to the “Actor training” block in the pseudocode of **Appendix F**, which shows this supervised optimization stage.
>
> - **Stage 2: Reasoning model compression (actor frozen).**
>
>     Once the actor is trained to understand latent messages, we **freeze the actor and communication adapter**, and train only the **reasoning model** to generate *compressed* latent communications.
>
>     This is described in **Section 3.3 Information Compression** and elaborated in **Appendix B.**
>
>     In this second stage, the actor provides a *fixed supervision signal* via its output distributions and latent feature geometry, forming the losses
>
>     ($\mathcal{L}_{\text{task}}, \mathcal{L}_{\text{pref}}, \mathcal{L}_{\text{geom}}$) (Eq. B.1).
>
>     Thus, the actor is *frozen* during compression training, ensuring semantic consistency and stable supervision.
>
> - **Why two stages.**
>
>     The separation improves stability and interpretability:
>
>     1. Stage 1 equips the actor with the ability to comprehend latent communication.
>     2. Stage 2 compresses reasoning outputs without altering the actor’s decoding behavior.
>
>         This hierarchical setup avoids co-adaptation and aligns with our stated goal of *studying communication efficiency rather than policy learning*.

---

> ### Author Response · Authors · 2025-11-15
>
> ### **(e) “Training-free” vs. “trained” settings**
>
> The terms *“training-free”* and *“trained”* refer to two complementary evaluation settings for analyzing **information compression**, introduced in **line 377** and defined more explicitly in **Table 2**.
>
> We clarify them below and will make their definitions explicit earlier in the revised version.
>
> - **Training-free setting.**
>
>     This setting evaluates the *intrinsic compressibility* of existing latent representations **without any additional training**.
>
>     Specifically, we take the latent communications generated by the pretrained reasoning model (Qwen2.5-Instruct) and **truncate or subsample** them at different retained ratios ($R \in [0,1]$) (e.g., 50%, 20%, 8 latent states) before feeding them to the actor.
>
>     No parameters are updated — this measures *how much task information survives when we directly shorten latent sequences*.
>
> - **Trained setting.**
>
>     In contrast, this setting corresponds to the **compression training stage** introduced in **Section 3.3 (lines 263–269)**.
>
>     Here, we train the reasoning model ($M_\phi$) to *actively produce shorter latent sequences* ($H_K$) that remain interpretable by a **frozen actor**, using the compression loss.

---

> ### Author Response · Authors · 2025-11-15
>
> ### Question
>
> **Question 1: How were the MHA and Projector trained?**
>
> The **Multi-Head Attention (MHA)** and **Projector** modules are *jointly trained as lightweight communication adapters* in the **first-stage actor training**, and kept **frozen during the second-stage compression training**.
>
> - **Where defined and used.**
>
>     The MHA + Projector form the *communication adapter* described in **Section 3.1 “Latent Communication” (lines 162–215)**.
>
>     As stated there, the latent states ($H={h_1,\dots,h_L}$) transmitted between agents are “processed by a trainable light-weight self-attention and a projection layer as a communication adapter for magnitude rescaling and helping the agent better interpret the latent meaning.”
>
> - **Training stage.**
>
>     During **Stage 1 (actor training)** described in **Section 3.2** and **Appendix C.1 (lines 755–809)** , the MHA + Projector parameters are **learned jointly with the actor** using the total loss. These gradients flow through the adapter, enabling it to rescale latent magnitudes and align representation between the sender and receiver.
>
> - **Freezing stage.**
>
>     Once the actor learns to interpret latent messages, both the **actor and its communication adapter (MHA + Projector)** are **frozen** in **Stage 2 (compression training)** (**Section 3.3 (lines 263–269)** and **Appendix B (lines 648–755))**.
>
> - **Summary of roles.**
>
>
>     | Component | Training Stage | Objective | Gradient Flow |
>     | --- | --- | --- | --- |
>     | MHA + Projector (adapter) | Stage 1: Actor training | Learn to normalize and interpret latent representations | Updated jointly with actor |
>     | MHA + Projector | Stage 2: Compression | Provide fixed mapping for compressed latents | **Frozen (no update)** |
>
> Thus, the **MHA and Projector are trained end-to-end with the actor in Stage 1** to enable latent-space alignment and are **frozen thereafter** to maintain a consistent communication interface during reasoning-model compression.
>
> ---
>
> **Question 2: There seems to be an assumption that the fixed actor can understand the latent communication of the reasoning model because they have the same backbone (e.g., Qwen2.5). Would the findings reported in the paper still hold if these two modules were based on different models?**
>
> Indeed, in our current experiments both the **reasoning (sender)** and **actor (receiver)** modules are based on the *same model backbone* (Qwen2.5-Base). This design choice was deliberate to **isolate the effect of latent-space communication itself**, rather than introduce additional confounds from architecture or capacity differences.
>
> We acknowledge that extending this framework to **cross-family settings** (i.e., sender and receiver models from different architectures) is a promising direction. The framework itself does **not require weight sharing** or identical backbones. A straightforward adaptation involves **adjusting the projector’s input and output dimensions** to match those of the respective sender and receiver models (e.g., LLaMA→Qwen).
>
> We consider this an important avenue for future work and plan to explore it systematically.

---

> ### Comment · Reviewer_1RDo · 2025-11-18
> **General response**
>
> I thank the authors for their detailed rebuttal. I will provide my general response here and follow up with specific questions in separate threads.
>
> First, I want to acknowledge the value of the core idea: enabling LLM agents to communicate via a hidden state. This effectively bridges the Comm-MARL and Multi-Agent LLM communities and offers promising research directions. Additionally, the experimental evaluation is solid given the complex computational framework.
>
> However, my main concern remains that the current presentation limits the paper's accessibility to the broader research community. While the authors pointed out in the rebuttal that the "missing information" technically exists in the text, there is a distinction between information being present and being digestible. For instance, the method descriptions are either being highly abstract (e.g., Line 216 regarding latent space) or over-detailed regarding implementation (e.g., Line 255 on hyperparameters), often lacking the necessary context to explain the design motivations.
>
> I strongly encourage the authors to incorporate the clarifications provided in the rebuttal into the main text. If a revision significantly improves the presentation and addresses the ambuiguity issues raised in my original review, I would be happy to raise my score and advocate for acceptance.

---

> ### Author Response · Authors · 2025-11-19
>
> We sincerely thank the reviewer for your constructive and encouraging feedback. We fully agree that **clarity and accessibility are essential** for the broader community to appreciate and build upon our work.
>
> In response to this concern, **we have incorporated all clarifications from our rebuttal directly into the main text** of the revised manuscript. For example:
>
> - A new related work that better position Interlat among previous work.
> - **Section 3.2 (Training Procedure)** now explicitly states that our training is *entirely offline and supervised* (no RL or environment reward gradients). It also provides a clearer account of how auxiliary components, such as the natural-language CoT plans, are obtained and used during training.
> - **Section 3.3 (Information Compression)** has been substantially rewritten to move key details from Appendix B into the main body, including: (i) the role of the *frozen instruction-tuned teacher model*, (ii) the *autoregressive latent-space generation mechanism*, and (iii) a **clear, self-contained explanation of the composite compression loss** with defined symbols ($w_t$, $\bar{z}$).
> - **Section 4.2 (Compression)** now **defines “training-free” and “trained” settings upfront** before their use, eliminating ambiguity.
> - All technical operations such as *stochastic replacement in curriculum learning* and *permutation for mismatched latents* are now described with **precise, implementation-level clarity**.
> - We have also **added a new qualitative analysis (Section 5)** with t-SNE visualizations to enhance the interpretability of latent communications
>
> Thank you again for your thoughtful review and for recognizing the potential of our work. We remain fully committed to addressing any further questions or concerns you may have.

---

> ### Author Response · Authors · 2025-11-26
>
> Dear reviewer 1RDo,
>
> We are eager to hear your thoughts on the improvements made and hope our revisions align more closely with your expectations. Your valuable feedback has been integral to this progress. As we approach the final stages of evaluation, your insights remain crucial. We kindly request your review of the latest version at your earliest convenience. Your expertise is greatly appreciated and will be instrumental in determining the final decision on our work. We look forward to your response with great anticipation.

---

### Official Review · Reviewer_z75q · 2025-10-31

**Soundness:** 2
**Presentation:** 2
**Contribution:** 2
**Rating:** 2
**Confidence:** 3

**Summary:**

This paper proposes Interlat, a framework that enables large language model (LLM)–based agents to communicate entirely in latent space instead of through discrete tokens. Interlat allows agents to transmit their final-layer hidden states as communication messages, processed through a lightweight adapter. The framework introduces additional training signals, including separation and alignment losses and employs curriculum learning to help the receiver agent gradually interpret latent messages. A second stage further compresses the latent communication to achieve more efficient reasoning while maintaining performance. Experiments on the ALFWorld environment compare Interlat against both no-communication and text-based baselines, as well as multiple ablations, demonstrating that latent communication improves task success rates and efficiency.

**Strengths:**

1. The idea of exchanging continuous latent representations rather than discrete tokens is conceptually novel and connects to how humans may communicate through visual or implicit signals rather than purely language.

2. The framework integrates several thoughtful components (JS separation loss, plan-alignment loss, curriculum learning) to ensure interpretable and robust communication. The compression analysis is particularly interesting, showing that latent messages can be shortened with minimal performance degradation.

**Weaknesses:**

1. Motivation and significance not fully articulated: While the paper demonstrates performance gains, it remains unclear what new behaviors or communicative properties latent communication enables beyond efficiency. For example, could this approach lead to more human-like communicative phenomena (e.g., implicit alignment, compositionality, or emergence of shared latent codes)? What is their grounding for semantic information being transmitted?

2. The paper should more explicitly position itself relative to existing literature, such as Learning to Communicate with Deep Multi-Agent Reinforcement Learning (Foerster et al., 2016) and other differentiable message-passing or embedding-based communication methods. Clarifying how Interlat differs conceptually and empirically would strengthen the contribution.

3. Some key terms lack clear definitions:
- What constitutes a supervised position and how are these positions selected?
- What exactly is the language space plan in the alignment loss?

4. The compression mechanism is one of the paper’s most promising contributions but is described briefly. The main text should include the loss formulation (currently only in Appendix B) and discuss how it maintains semantic alignment during compression. Referring to Table 3 without explaining these losses makes the narrative incomplete.

5. The task setup (ALFWorld) and evaluation metrics should be explained more clearly.
- What does accuracy or success rate precisely measure in these sequential tasks?
- What does step refer to — environment steps or rounds of inter-agent communication?

6. Providing qualitative examples or visualizations of the latent communication (e.g., probing or dimensional reduction) would make the results more interpretable.

**Questions:**

Since the curriculum gradually replaces text embeddings with latent states, the results seem to suggest that the text representations are initially easier for the receiver to interpret. If that is the case, it is unclear why latent-space communication is ultimately preferable. Would a simpler alternative yield comparable performance such as training the sender to regress the language-space plan?

---

> ### Author Response · Authors · 2025-11-15
>
> Thank you for your thorough review and feedback. Below is a detailed point-by-point response addressing your main concerns. **Please refer to the newly uploaded revised version of the paper to locate the corresponding line numbers.**
>
> ---
>
> First of all, we want to make some explaination around your concerns raised in weaknesses section.
>
> ---
>
> ### **Weakness 1: Response to “Motivation and significance not fully articulated”**
>
> Our work aims to go beyond simple speedup and uncover how **latent communication reshapes the behavioral and semantic dynamics of multi-agent systems**.
>
> **1. Emergent Behavioral Properties: Implicit Alignment and Exploration**
>
> Latent communication enables *implicit alignment* between agents by allowing them to exchange their internal hidden-state representations directly, instead of tokenized linguistic summaries. Empirically (Table 1, Fig. 3), this yields qualitatively different behaviors: agents trained with Interlat exhibit **longer yet more successful trajectories and more exploratory behavior**, even without explicit exploration objectives. This pattern reflects **shared understanding of latent intentions** analogous to nonverbal alignment in human communication rather than surface-level imitation. The “aha moment” (Fig. 3) further shows that the actor develops an internal capacity to discriminate and interpret another agent’s latent messages, suggesting the emergence of **task-specific representational grounding** rather than rote response to superficial cues.
>
> **2. Compositional and Parallel Representation of Reasoning Paths**
>
> Section 4.2 and Figure 4 demonstrate that latent communication sustains **parallel reasoning paths** across time, as evidenced by stable top-k probability gaps and lower ($P_{50}(S_{10})$). This indicates that compressed latent messages retain multiple plausible reasoning branches simultaneously, which is an emergent **compositionality** at the representational level that language-based pipelines lose after token discretization. In this sense, Interlat supports not only efficiency but also a richer “conceptual bandwidth” that allows multiple partially overlapping reasoning hypotheses to coexist and interact within communication.
>
> **3. Grounding of Semantic Information**
>
> The semantic grounding of transmitted information is ensured by the *conditional separation* and *plan-alignment* objectives (Eq. 1, L sep and L align in Sec. 3.2). These jointly force the actor to (a) distinguish between matched and mismatched latent messages, and (b) align its behavior with explicit language-space plans. This guarantees that what is transmitted is not arbitrary activation noise but **semantically meaningful, task-relevant structure,** which was ****validated empirically by the severe performance drop when **cross-task, noised, or randomly rotated latents** are substituted (Table 1). The model thus learns to interpret the latent communication as a *semantically grounded message* about the sender’s reasoning trajectory.
>
> ---
>
> **Weakness 2:  The paper should more explicitly position itself relative to existing literature**
>
> We appreciate the reviewer’s suggestion and have clarified our distinction from *Learning to Communicate with Deep Multi-Agent Reinforcement Learning* (Foerster et al., 2016) and other differentiable message-passing methods.
>
> While Foerster et al. focus on learning low-dimensional communication signals in reinforcement-learning settings (RIAL/DIAL), **Interlat operates in a fundamentally different regime**—large-language-model (LLM) agents communicating through *their own hidden states*. Instead of optimizing discrete or small continuous messages via policy gradients, Interlat directly transmits the final-layer latent representations of an LLM sequence, enabling far richer expressivity (≈40 k bits / state vs ≈15 bits / token) and tight coupling between reasoning agents in complex textual environments such as ALFWorld.
>
> **Conceptual and Empirical Differences**
>
> - **Modality:** Interlat communicates *latent sequences*, not discrete or embedding-level messages.
> - **Scope:** Prior work targets small RL agents; Interlat targets *LLM-based collaborative reasoning* and long-horizon planning.
> - **Architecture:** No external communication head is trained—hidden states are inserted directly into the receiver’s transformer input with a light adapter, forming an end-to-end latent channel.
>
>     Empirically, this yields higher task success rates and stability while permitting up-to-24× compression without loss of performance.

---

> ### Author Response · Authors · 2025-11-15
>
> ### **Weakness 3:  Some key terms lack clear definitions.**
>
> 3.1: What constitutes a supervised position and how are these positions selected?
>
> In alfworld settings, the supervised position is the thought and action parts of the training data, which we have provided a training templete in Appendix I, line 995.
>
> 3.2: What exactly is the language space plan in the alignment loss?
>
> In the first training stage, the reasoning model still generates messages in the **language space**, producing both textual tokens and their corresponding hidden states. As shown in Figure 1, we collect these hidden states to form latent communications, while the **language-space plan** refers to the sequence of generated **text tokens** (i.e., the natural-language plan) that corresponds one-to-one with those latent states.
>
> In the alignment loss, this language-space plan serves as a **semantic reference**, ensuring that the latent communication conveys information consistent with the reasoning that would be expressed through natural language.
>
> ---
>
> ### **Weakness4:  “Compression mechanism is promising but underexplained.”**
>
>  In the original submission, we placed the full loss formulation and optimization details in **Appendix B (lines 716–780)** due to space constraints. In the revision, we **moved the key equations and explanations into Section 3.3 (Information Compression)** and explicitly discuss how the three loss components.
>
> The mechanism for maintaining semantic alignment during compression is described in Appendix B . The uncertainty-weighted agreement loss ($L_{\text{pref}}$) matches the token-level predictive distributions between compressed and full latent paths, ensuring behavioral and informational consistency. The latent-direction alignment loss ($L_{\text{geom}}$) preserves the global semantic geometry of actor-side representations, preventing compressed latents from drifting to semantically different regions. Together with the task utility term, these components ensure that compression preserves meaning rather than merely reducing length. In the revision, we moved these key formulations into the main text (Section 3.3) for clarity.
>
> ---
>
> ### **Weakness5: The task setup (ALFWorld) and evaluation metrics should be explained more clearly.**
>
> As described in Section 4.1, we adopt the **ALFWorld benchmark which is** a standard embodied reasoning environment where the agent must complete multi-step household tasks (e.g., “put a clean apple in the fridge”) via natural-language actions grounded in a simulator.
>
> Each reasoning model produces a sequence of textual commands that are executed in the ALFWorld environment until success or termination. In alfworld, step” refers to **environment interaction steps** — each corresponding to a full action issued by the reasoning model and executed in ALFWorld.
>
> A detailed description of the task setup, environment interface, and evaluation metrics has been included in the revised version (**lines 257–262**).

---

> ### Author Response · Authors · 2025-11-15
>
> ## Question:
>
> ### **Response to: “Why latent-space communication instead of regressing the language-space plan?”**
>
> We appreciate the reviewer’s question and clarify that the curriculum in **Interlat** performs **stochastic** (not gradual) replacement of text embeddings with latent representations . During early curriculum stages, the receiver indeed finds text embeddings easier to interpret—this is also reflected in our “aha moment” analysis. However, text-based communication serves only as an **initial scaffold** to introduce the model to latent-space interaction. As training proceeds, latent-space communication becomes preferable due to its higher **efficiency** and **expressivity**.
>
> **(1) Conceptual difference**
>
> The *language-space plan* (P) represents a symbolic reasoning trace that helps the actor align with task semantics without drifting from the language domain. In contrast, the *latent communication* (H) encodes the underlying hidden-state trajectory—capturing richer, sub-symbolic features such as contextual uncertainty, compositional blending, and parallel reasoning hypotheses. We provide a clearer description of P in the revision of our paper in `lines 183-187`.
>
> Training the sender to **regress the language plan** would collapse these representations into discrete linguistic tokens, losing much of this structural and relational information.
>
> **(2) Empirical evidence**
>
> As shown in *experiments*, direct latent-state communication consistently outperforms language-regression baselines in both stability and success rate, particularly under **compression** and **cross-task transfer** conditions. This indicates that the latent channel preserves semantically grounded internal reasoning even when textual plans are ambiguous or noisy.
>
> **(3) Practical advantage**
>
> Unlike language-space communication, latent exchange can be efficiently **compressed** without decoding or re-encoding tokens. As described in Table 2, this process yields up to a **24× reduction in latency** at 8-step latents while maintaining task-level success. Thus, latent-space communication is not only semantically richer but also computationally more efficient.

---

> ### Author Response · Authors · 2025-11-26
>
> Dear reviewer z75q,
>
> We are eager to hear your thoughts on the improvements made and hope our revisions align more closely with your expectations. Your valuable feedback has been integral to this progress. As we approach the final stages of evaluation, your insights remain crucial. We kindly request your review of the latest version at your earliest convenience. Your expertise is greatly appreciated and will be instrumental in determining the final decision on our work. We look forward to your response with great anticipation.

---

### Official Review · Reviewer_2nNX · 2025-11-02

**Soundness:** 2
**Presentation:** 2
**Contribution:** 3
**Rating:** 4
**Confidence:** 3

**Summary:**

The paper proposes Interlat, a method that lets LLM agents communicate via latent representations instead of text. By exchanging hidden states through a small adapter network, agents share richer, faster information. On the ALFWorld benchmark, Interlat outperforms text-based and chain-of-thought baselines, even with up to 24× compressed latent messages. Results show that latent-only communication enables efficient and expressive multi-agent collaboration.

**Strengths:**

Interlat introduces an interesting way for language model–based agents to communicate directly through latent representations instead of discrete text. This marks an interesting shift in how multi-agent systems can share information, avoiding some of the inefficiencies of text-based communication. The idea seems especially promising for multimodal scenarios, where agents work across different representational spaces.

The proposed approach shows efficiency gains, achieving large reductions in communication latency. This makes it a strong candidate for scaling multi-agent systems where bandwidth and response time really matter. That said, these benefits likely come at the cost of interpretability, since it’s much harder to understand or inspect what’s being passed around in latent space compared to language-based exchanges.

In addition, the paper’s clear visualizations and detailed methodological explanations greatly enhance its readability and accessibility.

**Weaknesses:**

**Overstated and Misaligned Anthropomorphism**: The paper repeatedly compares latent-space communication to human *mind-reading*, which is wrong. What’s actually happening here is closer to the exchange of high-dimensional neural activations, not cognitive inference (or mind reading) on nonverbal cues as in human communication. This framing risks false anthropomorphizing what is essentially a representational alignment problem. I’d strongly recommend removing this metaphor to keep the paper conceptually grounded.

**Lack of Clarity Around Baselines**: The setup and training details for the baseline models are not clearly described. It’s hard to tell whether Interlat’s improvements are valid. The paper mentions that more details are in the Appendix, but that section is a copy-paste of the main text instead of providing the necessary experimental specifics. Without a clearer explanation, it’s difficult to judge how strong the improvements really are.

**Confounding from Shared Model Family**: All agents in the experiments use the same model backbone (Qwen2.5), which means the sender and receiver already share an aligned latent space. The observed performance gains might therefore reflect this intra-family compatibility rather than true inter-agent latent understanding. Testing communication across different model families or scales (e.g., Qwen -> LLaMA) would provide much stronger evidence for the claimed generality.

**Ambiguity in the “Information Parallel Budget” Analysis**: The "information parallel budget" idea is interesting but not very clearly defined or empirically grounded. The analysis treats differences in output probability distributions as evidence of parallel reasoning, but that interpretation is speculative — lower probability mass could just indicate more uncertainty, not necessarily parallel thought processes. Also, the analysis only looks at the first six hidden states, which likely misses longer-horizon reasoning effects. As it stands, this part feels underdeveloped and could use a clearer theoretical motivation or empirical validation.

**Questions:**

- Why do you use the base models as the actor models? Have you tried using instruction-tuned models?
- What is the actor model used for the Text baseline?
- In Table 1, why does Interlat require more steps than the other methods?
- Information parallel budget: Why do you use only the first six hidden states?
- How do you plan to address the interpretability issue? (I know this is out of scope, but I'm curious.)

---

> ### Author Response · Authors · 2025-11-15
>
> Thank you for your thorough review and feedback. Below is a detailed point-by-point response addressing your main concerns. **Please refer to the newly uploaded revised version of the paper to locate the corresponding line numbers.**
>
> ---
>
> First of all, we want to make some explaination around your concerns raised in weaknesses section.
>
> **Weakness 1: Overstated and Misaligned Anthropomorphism**:
>
> We thank the reviewer for pointing out this important clarification. Our use of “mind-reading” was metaphorical, inspired by *the functional similarity* between human non-verbal inference and direct latent exchange (`lines 44-47`), which is beyond language and better transimit our mind state to others.  We agree that our system does not perform same cognitive inference as in human mind-reading and we have propose the machine mind-reading only analogous but same to human mind-reading (`lines 44-47` and `lines 47-50`).
>
> ---
>
> ### **Weakness 2: Lack of Clarity Around Baselines**
>
> As detailed in **Appendix C (line 782)**, we have provided the **complete training configurations for all methods**, including both baselines and our proposed *Interlat*.  In particular, **lines 786–798** specify the exact training settings for the **CoT Plan** and **No Plan** baselines.
>
> The *Baselines* section is intended to clarify **which baselines and variants** are included in the comparison. For transparency and reproducibility, we additionally provide a **more technical mathematical formulation** in the Appendix than in the main text.
>
> Under these consistent configurations, we believe the **proposed method and all baselines are comparable**, and the reported improvements are valid.
>
> ---
>
> ### **Weakness 3: Confounding from Shared Model Family**
>
> Our primary objective in this work was to **first establish the intra-family feasibility** of *fully latent communication* under controlled settings where other confounding factors are minimized.
>
> We acknowledge that extending this framework to **cross-family settings** (i.e., sender and receiver models from different architectures) is a promising direction. A straightforward adaptation involves **adjusting the projector’s input and output dimensions** to match those of the respective sender and receiver models.
>
> We consider this an important avenue for future work and plan to explore it systematically.
>
> ---
>
> ### **Weakness 4: Ambiguity in the “Information Parallel Budget” Analysis**
>
> We thank the reviewer for this insightful comment and respectfully clarify that the *information parallel budget* is **not an anthropomorphic claim** about models “thinking in parallel.”
>
> Instead, it is a **quantitative, information-theoretic metric** that characterizes how *diverse reasoning hypotheses* are preserved in the model’s output distribution under latent-space compression.
>
> Our interpretation is grounded in three complementary perspectives:
>
> 1. **Probability dispersion reflects reasoning diversity, not mere uncertainty.**
>
>     In autoregressive transformers, the output probability distribution over the vocabulary indicates the number of *concurrent reasoning branches* that remain viable at each generation step. This experiment method has been empirically validated in *Coconuts* [2]. When a model collapses its latent representation into a single dominant trajectory, its top-k cumulative probability mass quickly concentrates (high P₅₀(S₁₀)). Conversely, when multiple reasoning paths are preserved, probability mass remains distributed across semantically coherent alternatives (stable top-k gaps). Thus, the **vertical gap between top-k bands** serves as a robust proxy for *representational parallelism* rather than generic prediction uncertainty.
>
> 2. **Empirical evidence supports this interpretation.**
>
>     As shown in **Figure 4**, trained reasoning models maintain **lower P₅₀(S₁₀)** alongside **higher success rates** and **lower cross-entropy** under compression (Table 2). This correlation suggests that broader probability support corresponds to *effective latent diversity,* i.e., the model retains multiple coherent reasoning hypotheses that the downstream actor can exploit.
>
>
> In the `original paper`, we further discuss this in greater depth, analyzing **32 hidden states as 32 reasoning steps**, and provide detailed comparisons of reasoning parallelism **before and after compression training** (`Appendix H, line 1051; Figure 9 in 1042; Figure 10 in 1129`).
>
> Hence, the *information parallel budget* constitutes a **principled, information-theoretic, and empirically validated** measure of **reasoning path diversity** in latent communication. Lower top-k concentration does not signify indecision but rather quantifies how effectively the model preserves multiple coherent reasoning hypotheses during compressed communication.

---

> ### Author Response · Authors · 2025-11-15
>
> ### **Question 1: Why do you use base models as the actor models? Have you tried using instruction-tuned models?**
>
> We intentionally use *base* models as actor models to **remove instruction-following biases** that are optimized for textual prompts. This design ensures that any observed improvement originates purely from **latent communication**, rather than from pretrained text-alignment priors.
>
> Moreover, *instruction-tuned* models (e.g., GPT-2, LLaMA 3.2-3B) have been **extensively adopted in prior work on latent-space studies** **[1, 2, 3]**. To maintain comparability with this literature, we align our setup with these studies. Beyond our new findings, we also observe results consistent with prior observations (`lines 866–869`), further validating our design choice.
>
> **References**
>
> **[1]** Shibo Hao, Sainbayar Sukhbaatar, DiJia Su, Xian Li, Zhiting Hu, Jason Weston, and Yuandong Tian. *Training Large Language Models to Reason in a Continuous Latent Space.* CoLM 2025.
>
> **[2]** Vignav Ramesh and Kenneth Li. *Communicating Activations Between Language Model Agents.* ICML 2025.
>
> **[3]** DiJia Su, Hanlin Zhu, Yingchen Xu, Jiantao Jiao, Yuandong Tian, and Qinqing Zheng. Token
> assorted: Mixing latent and text tokens for improved language model reasoning. In ICML 2025.
>
> ---
>
> ### **Question 2: What is the actor model used for the Text baseline?**
>
> As stated in **line 732**, the **same base model** is used for the *Text* baseline. It is trained directly in the **language space with identical training parameters**, ensuring a **fair and controlled comparison** with the latent communication setting.
>
> ---
>
> ### **Question 3: In Table 1, why does Interlat require more steps than the other methods?**
>
> As analyzed in the **main results (`line 311`)**, *Interlat* takes more steps because it **leverages multiple plausible reasoning paths** shared among agents through latent communication. This leads the actor model to engage in **more thorough exploratory behaviors**, resulting in longer trajectories but ultimately **better task performance**.
>
> A deeper analysis (`Table 1`, `Table 4`, `lines 881`) reveals a **nuanced relationship between step count and success rate**.
>
> While language-space actors tend to follow a single, text-guided plan—producing shorter but more rigid trajectories—their lower success rates indicate **limited exploration capacity**. They can become stuck or repeat suboptimal actions instead of exploring the environment effectively.
>
> Conversely, *Interlat*’s increased step count reflects **active exploration rather than inefficiency** with better performance.  However, more steps do not inherently guarantee better performance; as shown in *Table 4*, some settings take more steps yet achieve lower success rates.
>
> Together, these findings highlight that **step count must be interpreted jointly with success rate** to accurately assess an agent’s reasoning and problem-solving quality.
>
> ---
>
> ### **Question 4: Information Parallel Budget — Why do you use only the first six hidden states?**
>
> In the main text, we report results for **six hidden states** for brevity and readability.
>
> However, as noted in the **original version of the paper (`line 471`)**, we conducted a **more extensive analysis using 32 hidden states**, treating them as 32 reasoning steps to examine **parallelism before and after compression training**. Please see **Appendix H (`line 1048; Figure 9 at line 1042; Figure 10 at line 1129`)**, which was not inclued in main text due to space constraints.
>
> ---
>
> ### **Question 5: How do you plan to address the interpretability issue?**
>
> Thank you for the insightful question. We have directly addressed the interpretability of latent communication in our revised manuscript by providing qualitative visualizations via t-SNE in section 5 (`line 508`). This clustering provides strong empirical evidence that latent communications are not random noise but encode rich, task-specific semantic information.
>
> ---
>
> Hope these responses have clarified our work and addressed your concerns. We welcome any further questions you might have.

---

> ### Author Response · Authors · 2025-11-26
>
> Dear reviewer 2nNX,
>
> We are eager to hear your thoughts on the improvements made and hope our revisions align more closely with your expectations. Your valuable feedback has been integral to this progress. As we approach the final stages of evaluation, your insights remain crucial. We kindly request your review of the latest version at your earliest convenience. Your expertise is greatly appreciated and will be instrumental in determining the final decision on our work. We look forward to your response with great anticipation.

---

> > ### Comment · Reviewer_2nNX · 2025-11-26
> >
> > Thank you for the rebuttal!
> >
> > As I mentioned, this is an incorrect metaphor. Please explain why this would be analogous to human mind-reading or remove it from the paper. Human mind-reading does not involve interpreting brain signals. Please avoid using anthropomorphic metaphors that exaggerate the implications of this work. With many social media influencers now trying to attract attention through false or misleading claims from papers, it is important for us as researchers to stay grounded and rely on scientific, well-justified statements rather than misleading expressions.

---

> > > ### Author Response · Authors · 2025-11-27
> > >
> > > Thank you very much for this clarification and for highlighting the potential issues with our original wording. Your comment made us fully aware that our previous use of *“mind-reading”* can be misleading, and we completely agree that it is important for researchers to stay grounded and rely on precise, scientifically justified statements rather than anthropomorphic or attention-grabbing metaphors.
> > >
> > > After careful reconsideration, we have **removed the “mind-reading” metaphor from the paper**. Our original intention was not to suggest any interpretation of “brain signals”, but to convey the idea of **direct exchange of internal content without going through surface language**. To reflect this more accurately and avoid confusion with cognitive notions of human mind-reading, we now use the term **“telepathy” only as a loose, fictional analogy**, explicitly defined in the text as:
> > >
> > > > *“direct transmission of mental content between individuals without mediation by sensory channels or known physical mechanisms.”*
> > >
> > > In the revised version, we:
> > >
> > > * Replace *“mind-reading”* with this carefully defined notion of *telepathy* where appropriate,
> > > * Make it explicit that this is a **conceptual, fictional analogy** for *non-linguistic direct communication*, not a biological or neuroscientific claim, and
> > > * Substantially reduce such narrative or anthropomorphic language, keeping it only in **two places (abstract and introduction)** where it genuinely helps explain the high-level intuition that inspired our work.
> > >
> > >
> > > We also note that several recent works have begun exploring similar directions by using internal model states as mind/thought for richer inter-agent communication **[1,2,3]**, suggesting growing research interest and relevance in studying non-linguistic latent communication.  As one of the early efforts in this emerging area, we hope that the contribution of our method and the empirical insights demonstrated in the paper can be evaluated based on its technical novelty and measurable improvements in multi-agent coordination. We believe our extensive experiments and findings cab offer useful inspiration and groundwork for subsequent research in this space. While our motivation draws from speculative and science-fiction-inspired concepts **[4]**, our work is grounded in rigorous experimental validation and scientific methodology, aiming to contribute to the advancement of AI research. We view science fiction not as literal prediction, but as a creative lens that can help envision potential future directions and stimulate innovative frameworks for real-world system design.
> > >
> > > **[1]** Zheng Y, Zhao Z, Li Z, et al. Thought Communication in Multiagent Collaboration[J]. NeurIPS, 2025.
> > >
> > > **[2]** Fu T, Min Z, Zhang H, et al. Cache-to-Cache: Direct Semantic Communication Between Large Language Models[J]. arXiv preprint arXiv:2510.03215, 2025.
> > >
> > > **[3]** Zou J, Yang X, Qiu R, et al. Latent Collaboration in Multi-Agent Systems[J]. arXiv preprint arXiv:2511.20639, 2025.
> > >
> > > **[4]** Liu Cixin. The Dark Forest[J]. Tor Books, 2015.

---

### Note · Authors · 2026-01-04

I have read and agree with the venue's withdrawal policy on behalf of myself and my co-authors.